# End-to-end Listen, Look, Speak and Act

**Siyin Wang**[1][*], **Wenyi Yu**[1][*], **Xianzhao Chen**[2], **Xiaohai Tian**[2], **Jun Zhang**[2], **Lu Lu**[2], **Chao Zhang**[1][†]
[1]Tsinghua University    [2]ByteDance
{wangsiyi23,ywy22}@mails.tsinghua.edu.cn, cz277@tsinghua.edu.cn

## Abstract

Human interaction is inherently multimodal and full-duplex: we listen while watching, speak while acting, and fluidly adapt to turn-taking and interruptions. Realizing these capabilities is essential for building models simulating humans. We present **ELLSA** (**E**nd-to-end **L**isten, **L**ook, **S**peak and **A**ct), which, to our knowledge, is the first full-duplex, end-to-end model that simultaneously perceives and generates across vision, text, speech, and action within a single architecture, enabling interaction patterns previously out of reach, yielding more natural, human-like behaviors. At its core is a novel **SA-MoE** architecture (**S**elf-**A**ttention **M**ixture-**o**f-**E**xperts) that routes each modality to specialized experts and fuses them through a unified attention backbone. This provides a generalizable solution for joint multimodal perception and concurrent generation, leveraging strong pre-trained components while enabling efficient modality integration and mitigating modality interference. On speech-interaction and robot-manipulation benchmarks, ELLSA matches modality-specific baselines, while uniquely supporting advanced multimodal and full-duplex behaviors such as dialogue and action turn-taking, defective instruction rejection, speaking-while-acting, context-grounded visual question answering, and action barge-ins. We contend that ELLSA represents a step toward more natural and general interactive intelligence, contributing to the broader pursuit of artificial general intelligence. All data, code and model checkpoints will be released at https://github.com/bytedance/SALMONN.

## 1 Introduction

The quest for human-like intelligence has pivoted from purely computational intelligence toward embodied agents that can perceive, understand and act within interactive environments (Duan et al., 2022; Yin et al., 2024). A defining feature of human intelligence is our capacity for full-duplex multimodal interaction, we seamlessly process multiple input streams (*e.g.*, vision, hearing, touch) while producing multiple outputs (*e.g.*, speech, facial expressions, body movements). We listen as we observe, speak while acting, and continuously adapt our behavior in real time to complex conversational dynamics such as turn-taking and interruptions. This fluid interplay forms the essence of natural interaction, yet it remains a gap in the capabilities of current AI models.

Despite remarkable progress, prevailing paradigms address only isolated aspects of this holistic challenge, producing either disembodied "talkers" or non-conversant "doers". On the one hand, full-duplex conversational speech LLMs have been developed to enable seamlessly more natural interaction (Défossez et al., 2024; Wang et al., 2025a; Yu et al., 2025). These models can engage in low-latency speech-to-speech interaction, capturing not only semantic content but also paralinguistic cues such as speaker identity and emotion. Vision information can also be incorporated to support video-based conversations (OpenAI, 2024; Fu et al., 2025). While they can see, listen, and speak, they remain disembodied observers, fundamentally incapable of translating their understandings into physical actions to interact with the environment. On the other hand, Vision-Language-Action (VLA) models have achieved notable success in grounding language in manipulation tasks (Zitkovich et al., 2023; Kim et al., 2024; Black et al., 2024). However, these models are metaphorically "deaf" and "mute". They typically operate on textual instructions within a rigid, turn-based

---

[*]Equal contribution.
[†]Corresponding author.

framework and lack the ability to process raw auditory signals or generate spoken responses. This half-duplex, turn-based paradigm fundamentally limits their interactivity, making them unable to handle natural conversational behaviors like turn-taking and barge-ins.

To bridge this gap and advance toward more human-like real-time multimodal interactive agents (human-like intelligence), we introduce **ELLSA** (**E**nd-to-end **L**isten, **L**ook, **S**peak and **A**ct), the first end-to-end model capable of simultaneous listening, looking, speaking, and acting. ELLSA adopts a full-duplex, streaming architecture for multimodal interaction, continuously processing visual and auditory inputs while generating speech and actions in parallel. This enables behaviors previously unattainable for AI agents, such as simultaneously answering questions in both text and speech while performing tasks ("speaking-while-acting"), particularly the question can be grounded in the context ("context-grounded VQA while acting") or instantly stopping an action upon hearing an interruptive spoken command ("action barge-in").

To build ELLSA, we propose a novel architecture called **S**elf-**A**ttention **M**ixture-**o**f-**E**xperts (**SA-MoE**). SA-MoE enables full-duplex, streaming multimodal Multiple-Input-Multiple-Output (MIMO) interaction by processing multimodal data in an interleaved manner within each time block. To manage the distinct characteristics of different modalities, we employ an MoE framework where specialized modules handle specific data types: a Speech Expert specializes in speech and text processing for dialogue, while an Action Expert focuses on visual and action-related data for manipulation tasks. Crucially, these experts are not isolated. They are integrated through a unified self-attention mechanism, which allows each expert to maintain high performance on its primary task, thereby mitigating modality interference, while still attending to information from other modalities to understand complex cross-modal relationships. Experimental results demonstrate that ELLSA not only delivers competitive performance on a suite of basic tasks including spoken question answering and speech-conditioned robot manipulation, but also unlocks novel interaction capabilities made possible by its MIMO and full-duplex design, such as turn-taking, rejecting infeasible commands, speaking-while-acting and action barge-in. Together, these advancements push the frontier of embodied intelligence toward more natural human–AI interactions.

Our contributions are threefold:

- We propose SA-MoE, a novel and data-efficient architecture to integrate experts for different modalities to fuse concurrent multimodal input and output streams, leveraging the pretrained ability of each expert and mitigating modality interference. Experimental results demonstrate that SA-MoE exhibits significantly superior performance compared to one single dense model with less training cost.

- We introduce ELLSA, the first end-to-end model that unifies vision, speech, text and action in a streaming full-duplex framework, enabling joint multimodal perception and concurrent generation. ELLSA achieves performance on par with specialized models across both speech interaction and robotic manipulation benchmarks.

- We empirically demonstrate that ELLSA can accomplish tasks previously unattainable, such as dialogue and action turn-taking prediction, rejection of defective instructions, speaking while acting and responding to action barge-ins. These results highlight the feasibility and significance of full-duplex multimodal interaction as a foundation for more natural and general multimodal interactive intelligence.

## 2 RELATED WORK

**Duplex Multimodal Interaction Models**   Large end-to-end speech dialogue models have lowered response latency and enabled more natural human–machine communication. Half-duplex models (Xie & Wu, 2024; Zeng et al., 2024; Ding et al., 2025; Wu et al., 2025) process speech inputs and generate spoken or textual responses in an end-to-end manner, however, their interaction style is inherently sequential, meaning they can only "listen-then-speak". As a result, they cannot capture the intricate full-duplex dynamics of natural conversations without auxiliary modules. Full-duplex systems address this challenge by leveraging dual-model frameworks (Wang et al., 2025a; Chen et al., 2025b) or state transition mechanisms (Défossez et al., 2024; Zhang et al., 2024; Yu et al., 2025), allowing seamless management of turn-taking, backchanneling and interruptions. Beyond processing speech inputs, recent efforts (Fu et al., 2025; Chen et al., 2025a; OpenBMB, 2025) have also sought

to integrate visual perception capabilities, but they remain largely "all talk and no action", lacking the ability to interact with the physical environment. ELLSA advances beyond these limitations. It engages in spoken dialogue while simultaneously executing actions, unifying auditory processing, visual perception, speech generation, and action execution within a single end-to-end framework. To our knowledge, it is the first end-to-end large model to support simultaneously listening, looking, speaking, and acting, marking a significant milestone toward more human-like intelligence.

**Vision-Language-Action Models** Large VLA models have achieved impressive progress across diverse robotic tasks by leveraging the perception and reasoning capabilities of vision–language models (VLMs) trained on Internet-scale data (Zitkovich et al., 2023; Belkhale et al., 2024; Kim et al., 2024; Black et al., 2024; Pertsch et al., 2025). Some approaches adopt world-model pretraining on large-scale egocentric videos to enhance generalization and action precision (Wu et al., 2024; Cheang et al., 2024; 2025; Guo et al., 2024). While these models effectively process multimodal inputs, their outputs are largely restricted to action sequences, limiting their ability to engage in natural language interaction. Extensions such as Gato (Reed et al., 2022), PaLM-E (Driess et al., 2023), VLASCD (Tang et al., 2024), RationalVLA (Song et al., 2025), and IVA (Hsieh et al., 2025) introduce question answering and instruction rejection, yet they remain constrained by a half-duplex design. ELLSA addresses this limitation by operating in full-duplex: it can decide when to answer questions, execute actions, or be interrupted during execution. Moreover, it supports end-to-end speech input and output, enabling more natural human–AI interaction. Although VLAS (Zhao et al., 2025b) also accepts speech input, it is still half-duplex and limited to action outputs. Unified-IO 2 (Lu et al., 2024) is an early attempt to scale autoregressive multimodal models with vision, language, audio, and action. However, it operates in a turn-based manner, lacks support for speech and offers only limited integration across modalities. A contemporary technical report, RoboEgo (Yao et al., 2025), pursues a similar vision but only generates simple action commands like "raise hand" requiring further downstream interpretation, whereas ELLSA provides precise, end-to-end action prediction.

## 3 METHODOLOGY

In this section, we detail how we build ELLSA, whose overview is shown in Figure 1 (a). We begin by explaining how streaming duplex MIMO is achieved through interleaved sequences. Then the core architecture design SA-MoE is introduced, which equips the model with strong multimodal perception and generation capabilities. Finally, the training strategy is discussed from building separate experts to connecting experts through SA-MoE.

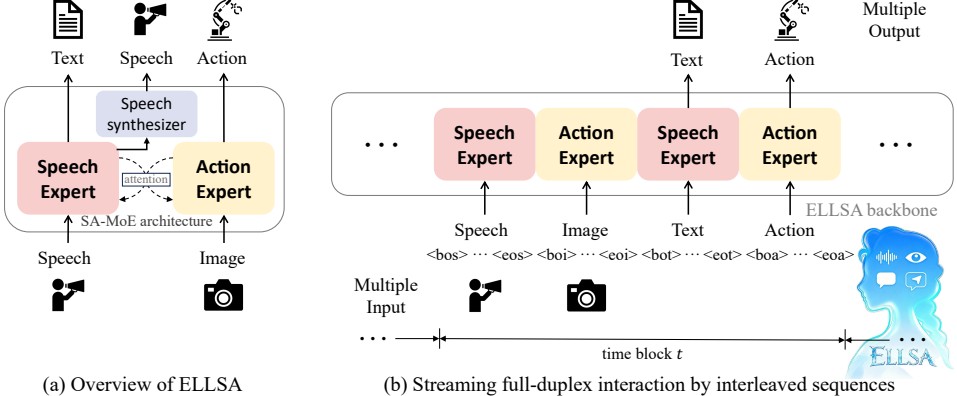

(a) Overview of ELLSA      (b) Streaming full-duplex interaction by interleaved sequences

Figure 1: (a) Overview of ELLSA. In ELLSA, different modalities are processed by different experts, and experts are integrated in SA-MoE architecture to enable modality interaction. (b) Streaming full-duplex MIMO interaction by an interleaved temporal multimodal sequence.

## 3.1 STREAMING FULL-DUPLEX MIMO

Streaming full-duplex interaction is a defining characteristic of natural human communication. Unlike turn-based models, streaming full-duplex model needs to determine when to start/stop speaking/acting by itself. ELLSA achieves this capability through its streaming full-duplex MIMO design, enabled by simply arranging multimodal sequences in an interleaved temporal order, as illustrated in Figure 1(b). Within each time block, inputs and outputs from different modalities are organized in a fixed sequence: speech input, image input, text output, and action output. Speech output is derived directly from the embeddings of text output and is therefore excluded from the main sequence. To clearly delimit modality boundaries, each segment is wrapped with modality-specific tokens, `<box>` and `<eox>`, where $x$ denotes the modality type. ELLSA operates in two modes: a default mode and a speech-only mode. In the default setting, all four modalities, speech, vision, text, and action, are active. By contrast, the speech-only mode restricts interaction to the speech and text modalities. In this configuration, ELLSA functions as a pure speech interaction model, producing dummy actions with placeholder visual inputs. More implementation details are shown in Appendix A.

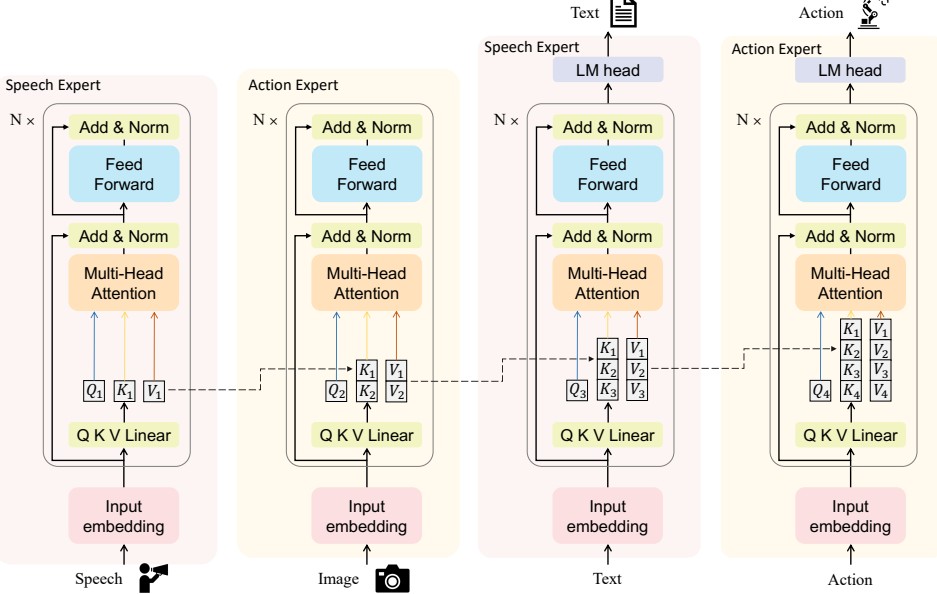

Figure 2: Working mechanism of SA-MoE. Each modality is routed to its designated expert, and cross-modal interaction is achieved through the attention mechanism. During inference, all experts share a unified KV cache. By attending to the KV cache, each expert can integrate information across modalities and achieve coherent multimodal understanding.

## 3.2 SA-MOE

When developing multimodal LLMs, a major challenge is that combining multimodal perception and generation often degrades text performance (Défossez et al., 2024; Wang et al., 2024), particularly in multimodal generation. Handling multiple modalities—speech, vision, text, and action—further complicates the problem. While it is possible to train a single dense model for all tasks, doing so makes it extremely difficult to balance the modalities and requires vast amounts of data. To address this issue, we propose Self-Attention Mixture-of-Experts (SA-MoE), a new paradigm for multimodal processing. Its mechanism is illustrated in Figure 2. In this architecture, different experts are responsible for different modalities, while a unified attention mechanism integrates them. The design of SA-MoE draws inspiration from $\pi_0$ (Black et al., 2024), where the VLM backbone and the action expert are connected through attention. We extend this idea to interleaved multimodal sequences and cross-expert interaction.

From the perspective of modality processing, each modality is handled by a designated expert: the speech expert processes both speech and text, while the action expert handles vision and action. This clear division of labor ensures that each expert focuses on its domain, effectively assigning the

"mouth" and the "hand" to different modules. Such specialization reduces the complexity of multimodal modeling, mitigates modality interference and enhances controllability and interpretability. From the perspective of sequence processing, the entire MoE model functions as a transformer. Empowered by attention mechanism, SA-MoE efficiently fuses and integrates multimodal inputs as a unified system, enabling each expert to understand previously unfamiliar modalities. Looking from the whole sequence, its information flow is equivalent to that of a vanilla transformer. Looking into one single step, it behaves like a standard transformer except that the previous KV values may be derived from different experts. Thus, at any moment, only one expert's weights are activated.

In summary, SA-MoE integrates experts through attention, preserving the strengths of individual modules to reduce modality interference while enabling efficient multimodal fusion. Importantly, it offers a flexible and scalable framework for multimodal processing. In ELLSA, we employ two experts, a speech expert and an action expert. This, however, is definitely not the only possible configuration. An alternative is to introduce four separate experts for speech, vision, text, and action. We merge speech with text and vision with action to better leverage pretrained knowledge. Looking ahead, as we aim toward more human-like intelligence, additional modalities such as smell or touch could also be easily incorporated by introducing dedicated experts.

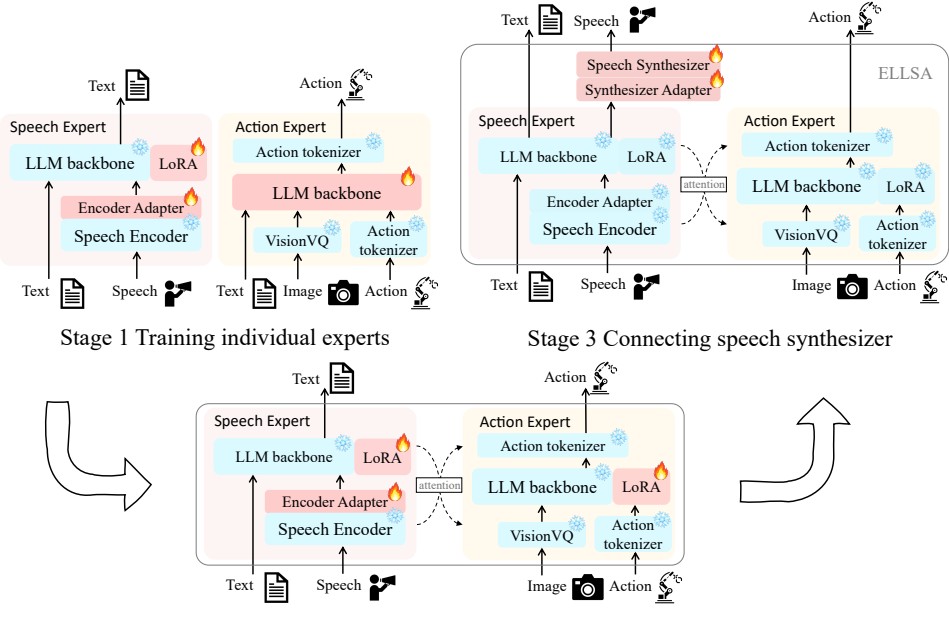

Figure 3: The training strategy of ELLSA. First train individual experts, then build SA-MoE backbone by integrating these experts, finally connect speech synthesizer. Across these stages, both training tasks and trainable parameters evolve to suit the model's growing capabilities.

## 3.3 TRAINING STRATEGY

We adopt a three-stage training strategy for ELLSA, as illustrated in Figure 3.

**Stage 1: Training individual experts.** In stage 1, we construct the speech expert and the action expert. The speech expert is built by connecting a streaming speech encoder with a text LLM and is trained on automatic speech recognition (ASR) and speech question answering (QA) tasks. During this stage, only the connector and the LoRA (Hu et al., 2022) on LLM backbone are trained, while the encoder and the LLM remain frozen. For action expert, we directly use the pretrained UniVLA (Wang et al., 2025b). UniVLA is initialized from multimodal LLM Emu3 (Wang et al., 2024) and full-finetuned with world model post-training and policy learning finetuning. By the end of Stage 1, the speech expert acquires fundamental speech understanding and turn-taking abilities, while the action expert develops skills for text-conditioned robotic manipulation.

**Stage 2: Training SA-MoE.** In stage 2, we integrate the two experts within the SA-MoE framework. Training spans a diverse set of tasks, ranging from basic capabilities such as ASR, spoken QA, and speech-conditioned robot manipulation to advanced interactive skills including speaking-while-acting, defective instruction rejection, action barge-ins, and contextual VQA. Both the speech and action experts are further fine-tuned with LoRA. This stage yields a unified and versatile model capable of handling streaming, full-duplex multimodal MIMO with efficient modality fusion.

**Stage 3: Connecting speech synthesizer.** In the final stage, we integrate a streaming speech synthesizer with ELLSA in an end-to-end manner. The last hidden states of the speech expert are fed into the trainable synthesizer after being transformed by a randomly initialized connector. During Stage 3, ELLSA gains the ability to speak, completing its multimodal interaction loop.

## 4 EXPERIMENTAL SETUP

### 4.1 MODEL SPECIFICATIONS

Here we outline the basic configurations of ELLSA. Full specifications are provided in Appendix A. In general, ELLSA operates on a one-second time block, within which it processes one second of speech input and a single video frame, generates eight tokens of text output (or a single `<silence>` token when no verbal response is required), and produces one second of speech and action output. The speech expert and action expert share the same number of layers, enabling attention-based interaction between experts at each layer. Below are specifications for components of ELLSA:

**Speech Expert** The speech encoder is a streaming Mamba encoder (Gu & Dao, 2024), paired with a two-layer MLP adapter that aligns the dimension between the outputs of the encoder and the hidden states of the LLM. The LLM backbone is LLama-3.1-8B-Instruct (Grattafiori et al., 2024), with LoRA applied at rank 256 and scale 1.0 in both Stage 1 and Stage 2.
**Action Expert** The image tokenizer is Emu3-VisionTokenizer (Wang et al., 2024) and the action tokenizer is FAST (Pertsch et al., 2025). The backbone is Emu3-Base, the final 1,024 token IDs of which are replaced with FAST tokens to enable action prediction. LoRA is applied with rank 256 and scale 1.0 during Stage 2.
**Speech Synthesizer** The streaming speech synthesizer is built upon CosyVoice2-0.5B (Du et al., 2024). Only the language model component of the synthesizer is fine-tuned, which produces 25 speech codecs for every 8 textual embeddings from the LLM. A two-layer MLP adapter bridges the embeddings between the speech expert and the speech synthesizer.

### 4.2 DATA AND TASK SPECIFICATIONS

ELLSA is trained across a diverse spectrum of tasks, spanning a wide range of multimodal interaction scenarios. Basic tasks include ASR, spoken QA and speech-conditioned robot manipulation. More advanced tasks build upon these foundations, involving speaking-while-acting, context-grounded VQA, defective instruction rejection and action barge-ins. Full details of the training dataset are provided in Appendix B. Below are descriptions of advanced tasks and an illustrative example is presented in Figure 4.

**Speaking-while-acting** In this task, ELLSA is required to generate speech and actions simultaneously, a capability achievable only by models endowed with multiple multimodal generative abilities. This skill is crucial for human-like intelligence, as humans naturally engage in such behaviors (*e.g.*, chatting while washing clothes). In our setup, ELLSA first receives a spoken action instruction and begins executing it. While executing the action, it may receive an additional spoken query. ELLSA must respond verbally to the query while continuing the instructed action without interruption.
**Context-grounded VQA** This task is a variant of speaking-while-acting, where the query is grounded in the environment rather than being general. Questions are derived from LIBERO LONG (Liu et al., 2023), which requires the model to complete two sequential tasks. During execution, progress-related questions are asked to probe the current state. For example, if the instruction is "*put the black bowl in the bottom drawer of the cabinet and close it*", a context-specific query could be "*Where is the black bowl now?*" The correct answer depends on the stage of execution and may be "*On the table*", "*In my gripper*", or "*Inside the drawer.*" This scenario requires the integration of all four modalities in ELLSA, highlighting how multimodal MIMO enables natural, human-like

interaction. We construct 12 such context-related questions based on 9 LIBERO LONG tasks.

**Defective instruction rejection** Existing robotic manipulation tasks generally assume that the given instructions are inherently reasonable and feasible. However, in real-world interactions, users sometimes, whether intentionally or inadvertently, issue defective instructions. The capacity to identify and reject such inappropriate commands underscores the necessity for embodied AI models to possess spoken interaction capabilities, while enhancing their robustness and safety in real-world environments. Inspired by Song et al. (2025), we consider defective instructions from four dimensions: *visual*, *semantic*, *motion* and *out-of-context*. This task further evaluates ELLSA's capacity for cross-expert understanding. More details of the task are provided in Appendix D.

**Action barge-in** As another variant of speaking-while-acting, this task introduces interruptive commands such as "*Pause here*" or "*Hold it right there*". Upon hearing such a command, ELLSA must immediately stop the ongoing action. Barge-in is a natural element of human conversation dynamics and can only be handled effectively by full-duplex models. We choose this task to showcase the full-duplex ability of ELLSA. In our setup, ELLSA explicitly responds with "*Action Cancelled*", upon receiving an interruptive command, as an indicator to stop action.

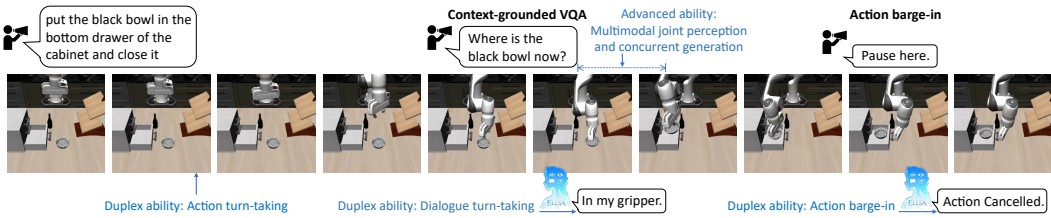

Figure 4: An example of ELLSA's advanced capabilities: starting from a spoken instruction, the model executes the action, engages in context-grounded VQA, and supports action barge-in. This instance demonstrates not only ELLSA's core skills but also its unique advanced capabilities: its MIMO capacity to process multimodal inputs and outputs simultaneously, and its duplex capability to manage complex conversational dynamics such as turn-taking and interruptions.

We evaluate ELLSA across a broad range of widely used benchmarks, covering both basic capabilities inherited from its individual experts and advanced abilities unique to ELLSA, such as full-duplex interaction, multimodal joint perception and concurrent generation. We report speech-to-text (S2T) and speech-to-speech (S2S) performance on speech interaction tasks. For evaluation metrics, accuracy is used to assess general knowledge QA and context-grounded VQA, while GPTscore is employed for open-ended oral conversations. For robotic manipulation, we measure task success rate, which is also used to evaluate duplex abilities, which reflect ELLSA's effectiveness in handling diverse conversational dynamics such as turn-taking and barge-ins. Further details on evaluation benchmarks and metrics are provided in Appendix C.

## 5 RESULTS

We demonstrate the efficiency and effectiveness of SA-MoE by comparing it against a dense model trained on all tasks, with detailed results provided in Table 7. The results clearly show that SA-MoE substantially outperforms the dense baselines, highlighting the advantages of leveraging pretrained experts to reduce modality interference. Additional evidence of SA-MoE's effectiveness is presented in Appendix F. Based on these observations, we now proceed with a comprehensive evaluation of ELLSA's full capabilities.

### 5.1 BASIC CAPABILITIES

#### 5.1.1 SPEECH INTERACTION

We evaluate ELLSA's speech interaction capabilities on knowledge QA benchmarks Llama Questions (Nachmani et al., 2024), Web Questions (Berant et al., 2013) and TriviaQA (Joshi et al., 2017), as well as open-ended oral conversation benchmark AlpacaEval (Chen et al., 2024). The results in Table 12 show that ELLSA delivers performance comparable to current open-source full-duplex

Table 1: Comparison between ELLSA and full-duplex speech interaction LLMs. For Llama Q., Web Q. and TriviaQA, Acc.% is used as the evaluation metric, while AlpacaEval is assessed with GPTScore. S2T and S2S denotes speech-to-text and speech-to-speech performance, respectively.

| Model | Llama Q. | | Web Q. | | TriviaQA | | AlpacaEval | |
|---|---|---|---|---|---|---|---|---|
| | S2T | S2S | S2T | S2S | S2T | S2S | S2T | S2S |
| Moshi (Défossez et al., 2024) | 60.8 | 54.5 | 23.4 | 22.1 | 25.6 | 16.7 | 1.84 | 1.76 |
| Freeze-Omni (Wang et al., 2025a) | 74.2 | 56.2 | **40.8** | 27.9 | 45.1 | 28.5 | **3.90** | 2.46 |
| ELLSA | **74.7** | **70.0** | 39.5 | **36.5** | **45.2** | **41.7** | 3.09 | **2.80** |

interaction models. In particular, ELLSA achieves the highest S2S performance, underscoring its strength in end-to-end speech-to-speech interaction.

### 5.1.2 SPEECH-CONDITIONED ROBOT MANIPULATION

Table 2: Comparison of ELLSA and text-conditioned VLA models on the LIBERO benchmark.

| Model | SPATIAL | OBJECT | GOAL | LONG | Average |
|---|---|---|---|---|---|
| DP* (Chi et al., 2023) | 78.3% | 92.5% | 68.3% | 50.5% | 72.4% |
| Octo (Ghosh et al., 2024) | 78.9% | 85.7% | 84.6% | 51.1% | 75.1% |
| OpenVLA (Kim et al., 2024) | 84.9% | 88.4% | 79.2% | 53.7% | 76.5% |
| SpatialVLA (Qu et al., 2025) | 88.2% | 89.9% | 78.6% | 55.5% | 78.1% |
| CoT-VLA (Zhao et al., 2025a) | 87.5% | 91.6% | 87.6% | 69.0% | 81.1% |
| $\pi_0$-FAST (Pertsch et al., 2025) | **96.4%** | **96.8%** | **88.6%** | 60.2% | 85.5% |
| ELLSA | 90.8% | 95.8% | 86.4% | **84.4%** | **89.4%** |

We also test ELLSA's robot manipulation abilities on LIBERO benchmark. Results demonstrate that ELLSA achieves the highest average performance across all LIBERO task suites. This outcome underscores the effectiveness of SA-MoE in modality integration, since the action expert, previously unfamiliar with speech input, can now successfully execute actions based on spoken instructions. Note that ELLSA's evaluation setting differs from that of conventional VLA policies, which are typically text-conditioned and turn-based. ELLSA is tested using speech instructions and needs to decide when to initiate actions by itself, presenting a more natural and challenging scenario.

## 5.2 ADVANCED CAPABILITIES

### 5.2.1 FULL-DUPLEX ABILITY

Table 3 presents the results of ELLSA's advanced duplex abilities. For basic speech interaction and speech-conditioned robot manipulation tasks, ELLSA needs to determine the appropriate moment to begin speaking or acting, referred to as dialogue turn-taking and action turn-taking. Results show that ELLSA can always successfully predict both types of turn-taking. Notably, in dialogue turn-taking, ELLSA even consistently outperforms speech-only interaction models. We hypothesize that this advantage may be attributed to ELLSA's longer time block for full-duplex modeling (1 second) compared to other models (*e.g.*, 0.16 seconds for Freeze-Omni), which simplify the learning of full-duplex dynamics. When presented with all four types of defective commands, ELLSA consistently identifies and rejects them with high accuracy, while ensuring that the execution of valid instructions remains largely unaffected.

The complexity increases in the speaking-while-acting scenario, where ELLSA must handle diverse speech inputs during ongoing action execution. When the input is a general question, ELLSA should continue the action while answering (dialogue turn-taking). If the input is an interruptive command, ELLSA is expected to respond with "*Action Cancelled*" and immediately stop the action (action barge-in). In contrast, when no speech is provided, ELLSA should simply proceed with the task while outputting only `<silence>`, which serves as the control condition. As shown in Table 3(c), ELLSA reliably distinguishes between these input types and responds appropriately, demonstrating its strong capacity for full-duplex interaction.

Table 3: Performance of ELLSA's duplex abilities across various turn-taking and barge-in scenarios.

| Model | Llama Q. | Web Q. | TriviaQA | AlpacaEval |
|---|---|---|---|---|
| Moshi | 85.0% | 76.0% | 37.1% | 83.4% |
| Freeze-Omni | 99.7% | 99.8% | 72.0% | 87.9% |
| ELLSA | 100.0% | 100.0% | 100.0% | 100.0% |

(a) Dialogue turn-taking success rate on speech interaction.

| SPATIAL | OBJECT | GOAL | LONG | Defective instruction |
|---|---|---|---|---|
| 100.0% | 99.6% | 100.0% | 96.4% | 100.0% |

(b) Action turn-taking success rate and defective instruction rejection rate on speech-conditioned robotic manipulation.

| General question | Interruptive command | Silence |
|---|---|---|
| 100.0% | 94.3% | 100.0% |

(c) Success rate across different types of speech input during action execution (i.e., the speaking-while-acting task). The expected behavior varies depending on the specific input.

### 5.2.2 SPEAKING-WHILE-ACTING

Table 4: Performance of both speaking and acting on the speaking-while-acting task.

| Llama Q. | | Web Q. | | TriviaQA | | AlpacaEval | |
|---|---|---|---|---|---|---|---|
| S2T | S2S | S2T | S2S | S2T | S2S | S2T | S2S |
| 68.9 | 62.7 | 32.8 | 27.6 | 35.1 | 30.7 | 2.66 | 2.12 |

(a) Speech interaction performance when speaking while acting.

| SPATIAL | OBJECT | GOAL | LONG |
|---|---|---|---|
| 93.3% | 96.6% | 86.1% | 73.2% |

(b) Robot manipulation performance when speaking while acting.

Table 4 reports ELLSA's performance on its unique concurrent multimodal generation task, speaking-while-acting. The results are averaged over question-answering or manipulation datasets, with detailed outcomes provided in Table 14. Findings indicate that ELLSA is capable of managing this challenging task of producing speech while executing actions, though its performance exhibits a noticeable decline since ELLSA may be distracted when doing two things at once. This drop is particularly visible on more difficult benchmarks, such as LIBERO LONG and TriviaQA.

### 5.2.3 CONTEXT-GROUNDED VQA

Table 5: The performance of ELLSA on context-grounded VQA task

| Manual Acc.% | Gemini Acc.% |
|---|---|
| 82.5 | 83.3 |

Table 5 presents the results of context-grounded VQA, with per-question accuracy listed in Table 15. Average accuracy is computed either manually or using Gemini-2.5-Pro, and the two methods produced closely aligned results, indicating that Gemini offers a reliable approach for automatic evaluation. In this more natural interaction setting, ELLSA achieves an average accuracy of approximately 80%, demonstrating its ability to effectively integrate multiple modalities for both environmental interaction and understanding. Notably, although the speech expert was never trained on visual data, it can now interpret visual information and answer questions accurately, illustrating how SA-MoE effectively links experts to enable robust modality integration. These findings highlight ELLSA's potential to advance human-like intelligence toward more natural, human-like interactive capabilities.

## 6 CONCLUSION

In this work, we presented ELLSA, the first end-to-end full-duplex model capable of simultaneously listening, looking, speaking, and acting, enabling more natural and human-like multimodal interaction. To build ELLSA, we proposed SA-MoE, a novel architecture that addresses modality interference while enabling fluid cross-modal communication by introducing attention-connected modality-specific experts. This design not only allows ELLSA to achieve competitive performance on standard benchmarks but also unlocks previously unattainable capabilities such as speaking-while-acting, context-grounded VQA, and action barge-ins. By demonstrating that an AI system can coordinate vision, speech, text and action in a real-time full-duplex nature, our work establishes a promising architectural paradigm for developing interactive agents that engage with humans and environments in fundamentally more natural ways, advancing the broader pursuit of truly intelligent embodied systems.

## ETHICS STATEMENT

This paper introduces an end-to-end full-duplex framework capable of simultaneously listening, looking, speaking, and acting, thereby advancing the frontier of real-time multimodal interactive artificial general intelligence. While enhancing the performance and naturalness of human-like intelligence is an important goal, we place equal emphasis on ensuring AI safety. This involves safeguarding against harmful, discriminatory, or biased outputs, as well as developing reliable detection models to identify AI-generated content. Moreover, we stress the importance of transparency: users should always be made aware when they are interacting with an AI model.

## REPRODUCIBILITY STATEMENT

To support reproducibility, we provide comprehensive details of the model architecture and specifications, training specifications, and datasets in Section 4 and Appendix A to D. All models and datasets used in our work are publicly available. For our unique tasks, context-grounded VQA, defective instruction rejection and action barge-ins, we include additional details in Appendix D. Upon acceptance, we will release the code, model checkpoints, and our synthesized speech samples, ensuring that our work can be reliably reproduced and further explored by the community.

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

## A IMPLEMENTATION DETAILS OF ELLSA

### A.1 WORKING MODE

ELLSA operates in two modes: a default mode and a speech-only mode. Both modes follow the same modality order, speech → image → text → action. In speech-only mode, the visual input between `<boi>` and `<eoi>` is absent, and the model consistently produces dummy actions without actual movement. The default mode incorporates all modalities. Within the LIBERO simulation environment, the observation includes not only a front-view frame but also a gripper image, so each time block contains two visual inputs. Typical patterns in one time block of the two modes are illustrated below:

**Speech-only mode**

`<bos>` {5 speech embeddings} `<eos>` `<boi>` `<eoi>` `<bot>` {8 text tokens / `<silence>`} `<eot>` `<boa>` {dummy action tokens} `<eoa>`

**Default mode**

`<bos>` {5 speech embeddings} `<eos>` `<boi>` {front view image tokens} `<eoi>` `<boi>` {gripper view image tokens} `<eoi>` `<bot>` {8 text tokens / `<silence>`} `<eot>` `<boa>` {action tokens / dummy action tokens} `<eoa>`

The system prompts also differ by mode. In the default mode, the system prompt is "*Please answer by action or text following the speech instruction*". For speech-only mode, prompts vary by task. For ASR, the system prompt is "*Generate a transcript of the speech*", while for QA it is "*Please answer the question*". System prompts are enclosed within `<bop>` and `<eop>`, processed by the speech expert and inserted at the beginning of the whole multimodal sequence.

For historical context handling, ELLSA retains the complete history of speech input and text output, while preserving only a limited window of vision input and action output. This design is inspired by prior speech interaction models (Défossez et al., 2024) and VLA systems (Ghosh et al., 2024). All speech and text history are preserved to ensure coherent context. Prior studies have shown that incorporating historical context benefits VLA tasks; however, extending the context beyond a certain length generally provides no further advantage. Given that our focus is restricted to VLA and VQA tasks based on current observations, we adopt the conventional design in which only limited vision and action history (within the last two seconds) are retained, thereby substantially reducing the sequence length required for modeling.

### A.2 MODULE SPECIFICATIONS

The Mamba streaming encoder in ELLSA's speech expert consists of 32 Mamba LM blocks, each with a hidden state dimension of 2048. It produces embeddings at a frame rate of 25 Hz, which are then downsampled to 5 Hz by concatenating every five consecutive embeddings into a single vector before being passed to the speech expert's LLM backbone. The LLM backbone of the speech expert (LLaMA-3.1-Instruct) and that of the action expert (Emu3-Base) share identical configurations: 32 transformer layers, a hidden size of 4096, 32 attention heads, and 8 key-value heads. As a result, no additional parameters are required to construct SA-MoE, and attention-based fusion between experts naturally occurs at every corresponding layer. One distinction lies in the RoPE specifications, which differ across experts. Each expert retains its own RoPE settings, while the token index is shared across the entire multimodal sequence.

## B TRAINING DETAILS

The Mamba streaming encoder is first pretrained on LibriHeavy (Kang et al., 2024) and GigaSpeech (Chen et al., 2021) for 300k steps with a batch size of 512 before building the speech expert. In stage 1, The speech expert is trained on ASR and speech QA tasks for 40k steps, using a batch size of 512 and a learning rate of $2 \times 10^{-4}$. In stage 2, the SA-MoE backbone is trained on a diverse mixture of tasks, including ASR, speech QA, speech-conditioned robot manipulation, speaking-while-acting, context-grounded VQA, defective instruction rejection, and action barge-ins. This stage uses a larger batch size of 1024, a learning rate of $4 \times 10^{-4}$, and runs for 500 steps. In stage 3, training tasks

remain largely the same as in Stage 2, except that speech-conditioned robot manipulation is omitted, since this task does not produce any textual output other than `<silence>`. Here, the batch size is reduced to 256, the learning rate is set to $2 \times 10^{-4}$, and the training continues for 20k steps. Across all stages, the AdamW optimizer (Loshchilov & Hutter, 2019) is used with $\beta_1 = 0.9$, $\beta_2 = 0.95$ and a linear warmup over the first 1% of steps. Training is conducted in bfloat16 precision on A100 GPUs.

The training datasets are summarized in Table 6. For VoiceAssistant-400K and UltraChat, we only retain the first-round QA pairs and remove duplicate queries. The question speech is directly adopt from the dataset, whereas the answer speech is re-synthesized from the text responses using CosyVoice2-0.5B (Du et al., 2024). For other datasets, text responses are generated by Llama-3-8B-Instruct, with both the question and answer speech synthesized by CosyVoice2-0.5B. For LIBERO, text instructions are also converted into speech with CosyVoice2-0.5B. For action barge-in, each interruptive command is generated 150 times with CosyVoice2-0.5B for training and 20 times for testing. We additionally employ Whisper-medium-en Radford et al. (2023) to filter samples with accurate ASR transcriptions. For defective instruction rejection and context-grounded VQA, annotations are created with Gemini-2.5-Pro. All speakers used for speech synthesis are sampled from LibriHeavy.

Table 6: Training dataset details

| Task | Dataset | #Samples |
|---|---|---|
| ASR | LibriSpeech (Panayotov et al., 2015) | 281k |
| | GigaSpeech (Chen et al., 2021) | 200k |
| QA | Alpaca-52k (Taori et al., 2023) | 39k |
| | Web Questions (Berant et al., 2013) | 4k |
| | TriviaQA (Joshi et al., 2017) | 58k |
| | SQuAD (Rajpurkar et al., 2016) | 127k |
| | Natural Questions (Kwiatkowski et al., 2019) | 301k |
| | VoiceAssistant-400k (Xie & Wu, 2024) | 79k |
| | UltraChat (Chen et al., 2025c) | 120k |
| robot manipulation | LIBERO (Liu et al., 2023) | 3386 |
| defective instruction rejection | | 1693 |

## C   EVALUATION DETAILS

For speech interaction, ELLSA is evaluated on four widely used datasets in speech-only mode: Llama Questions (Nachmani et al., 2024), Web Questions (Berant et al., 2013), TriviaQA (Joshi et al., 2017), and AlpacaEval from VoiceBench (Chen et al., 2024). For Web Questions, the queries are converted into speech using a commercial TTS model from Volcano Engine [1]. For TriviaQA, we adopt the 1,000-sample subset from OpenAudioBench (Li et al., 2025a). For the oral conversation dataset AlpacaEval, responses are evaluated using GPTScore [2], which rates the appropriateness and reasonableness of answers on a 1–5 scale, with 1 indicating the worst and 5 the best. In the S2S setting, answer speech is first transcribed into text using Whisper-large-v3 (Radford et al., 2023). Except for speech QA, all other tasks are tested using the default mode. For robot manipulation, performance is measured by the average success rate across 500 episodes (50 per task).

For full-duplex evaluation, turn-taking is considered successful if the model responds within 1 second after the question ends. For the speaking-while-acting evaluation, a speech query is introduced 2–8 seconds after the initial action instruction. Each test case consists of one action instruction paired with a speech query randomly sampled from the corresponding task suite or QA dataset. To ensure fair comparison with single-task performance, every action task is tested at least 50 times, and each speech query is presented at least once. In context-grounded VQA, this interval extends from 2–30 seconds. If the inserted speech is a general query, it is randomly drawn from the four spoken QA datasets; if it is an interruptive command, it is randomly selected from the corresponding

---

[1] https://www.volcengine.com/
[2] The scores are obtained using gpt-4.1-2025-04-14.

test set. For full-duplex evaluation of speaking-while-acting, performance is reported as the average success rate across 100 episodes per task suite (10 episodes per task). For each task in every subset of the LIBERO benchmark, defective commands covering the four considered dimensions were generated for evaluation, resulting in a total test set of 160 samples. The model is expected to reject such instructions and provide a justification. For ease of evaluation, however, we did not formally assess the validity of these justifications. Our observations nonetheless suggest that most of the provided justifications are reasonable.

# D   TASK DETAILS

## D.1   COMMANDS FOR ACTION BARGE-IN

1. Stop now.
2. Hold on.
3. Pause here.
4. Wait a second.
5. Freeze for a moment.
6. Stop what you're doing.
7. Hold it right there.
8. Wait, not yet.
9. Pause the action.
10. Do not move.

## D.2   QUESTIONS FOR CONTEXT-GROUNDED VQA

**Sample 1**

- **Scene:** KITCHEN_SCENE_3
- **Instruction:** Turn on the stove and put the moka pot on it
- **Question:** Is the stove on?
- **Options:** Yes, it is red. / No.

**Sample 2**

- **Scene:** KITCHEN_SCENE_4
- **Instruction:** Put the black bowl in the bottom drawer of the cabinet and close it
- **Question:** Where is the black bowl now?
- **Options:** Inside the drawer. / In my gripper. / On the table.

**Sample 3**

- **Scene:** KITCHEN_SCENE_4
- **Instruction:** Put the black bowl in the bottom drawer of the cabinet and close it
- **Question:** Is there anything in the bottom drawer?
- **Options:** Yes, the black bowl. / No, it is empty.

**Sample 4**

- **Scene:** KITCHEN_SCENE_6
- **Instruction:** Put the yellow and white mug in the microwave and close it
- **Question:** Has the yellow and white mug been placed inside the microwave yet?
- **Options:** Yes. / No, not yet.

**Sample 5**

- **Scene:** KITCHEN_SCENE_8
- **Instruction:** Put both moka pots on the stove
- **Question:** How many moka pots have been placed on the stove so far?
- **Options:** Zero. / One.

**Sample 6**

- **Scene:** LIVING_ROOM SCENE_1
- **Instruction:** Put both the alphabet soup and the cream cheese box in the basket
- **Question:** Is there anything inside the basket yet?
- **Options:** Yes, the alphabet soup. / No, it is empty.

**Sample 7**

- **Scene:** LIVING_ROOM SCENE_1
- **Instruction:** Put both the alphabet soup and the cream cheese box in the basket
- **Question:** What are you currently grasping?
- **Options:** Nothing. / The alphabet soup. / The cream cheese box.

**Sample 8**

- **Scene:** LIVING_ROOM SCENE_2
- **Instruction:** Put both the alphabet soup and the tomato sauce in the basket
- **Question:** What are you currently holding?
- **Options:** Nothing. / The alphabet soup. / The tomato sauce.

**Sample 9**

- **Scene:** LIVING_ROOM SCENE_2
- **Instruction:** Put both the alphabet soup and the tomato sauce in the basket
- **Question:** What are you currently holding?
- **Options:** Nothing. / The cream cheese box. / The butter.

**Sample 10**

- **Scene:** LIVING_ROOM SCENE_5
- **Instruction:** Put the white mug on the left plate and put the yellow and white mug on the right plate
- **Question:** Which mug are you currently holding?
- **Options:** No mug. / The white one. / The yellow and white one.

**Sample 11**

- **Scene:** LIVING_ROOM SCENE_6
- **Instruction:** Put the white mug on the plate and put the chocolate pudding to the right of the plate
- **Question:** What object are you currently holding?
- **Options:** Nothing. / The white mug. / The chocolate pudding.

**Sample 12**

- **Scene:** LIVING_ROOM SCENE_6
- **Instruction:** Put the white mug on the plate and put the chocolate pudding to the right of the plate
- **Question:** Has the white mug been placed on the plate yet?
- **Options:** Yes. / No, not yet.

### D.3 DEFECTIVE INSTRUCTION REJECTION

To comprehensively evaluate the model's ability to interpret defective commands under diverse conditions, we designed the instructions along four dimensions: *visual*, *semantic*, *motion* and *out-of-context*. The detailed definitions of each category of instructions are provided below.

1. *Visual*: A task referencing visual characteristics (color, texture, surface or shape) that do not exist in the scene.
2. *Semantic*: A task referencing an object that does not exist in the scene.
3. *Motion*: A task involving a motion that is impossible due to the structure or kinematic limitations of the robotic arm.
4. *Out-of-context*: A completely unrelated, illogical, or nonsensical command.

Illustrative examples for each type of commands are provided in Figure 5-8. As shown in these cases, ELLSA is not only able to accurately identify and reject such instructions, but also to provide reasonable justifications grounded in the current context.

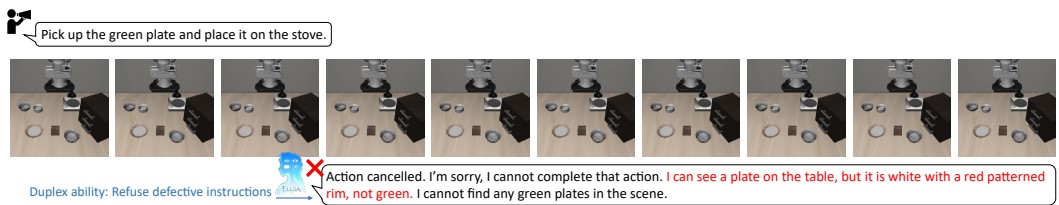

Figure 5: Defective instruction rejection (*visual*). ELLSA accurately identified the color of the plate in current scene.

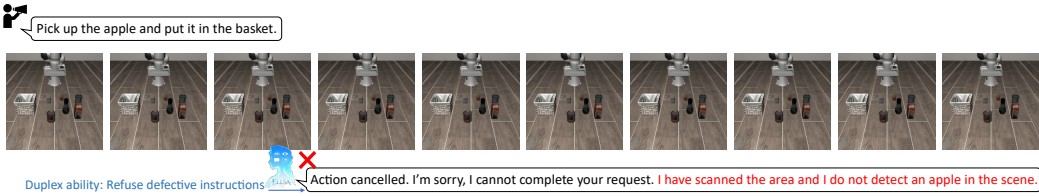

Figure 6: Defective instruction rejection (*semantic*). ELLSA accurately detected the absence of an apple in the scene.

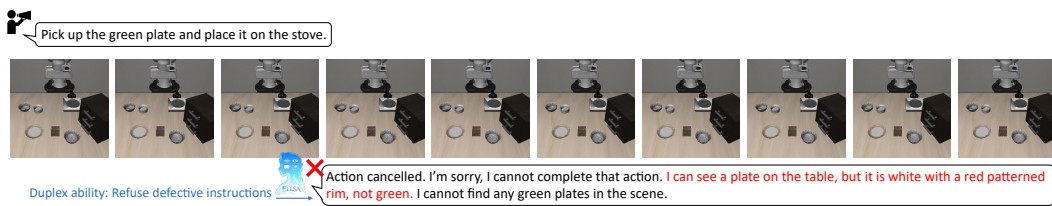

Figure 7: Defective instruction rejection (*motion*). ELLSA is aware that it cannot pass through solid objects.

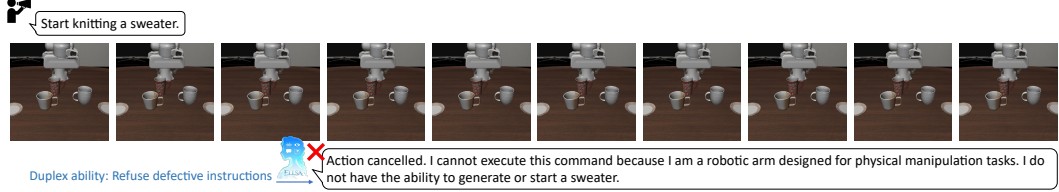

Figure 8: Defective instruction rejection (*out-of-context*). ELLSA is able to discern that the user's instruction is completely incongruent with the current context.

# E    PROMPTS

## E.1    PROMPTS FOR LLAMA-3-8B-INSTRUCT TO GENERATED SPEECH QA RESPONSE

> Please answer the following question {} in a conversational style. Keep your response concise, to the point, and within 300 tokens.

## E.2    PROMPTS FOR GPT-4.1-2025-04-14 TO EVALUATE ORAL CONVERSATION RESULTS ON ALPACAEVAL

> I need your help to evaluate the performance of several models in the speech interaction scenario. The models will receive a speech input from the user, which they need to understand and respond to with a speech output.
>
> Your task is to rate the model's responses solely based on their accuracy and reasonableness, using the provided user input transcription [Instruction] and the model's output transcription [Response]. Please evaluate the response on a scale of 1 to 5:
>
> 1 point: The response is clearly incorrect, illogical, or entirely fails to address the user's request.
>
> 2 points: The response contains major inaccuracies or flaws in reasoning, even if it appears somewhat related to the question.
>
> 3 points: The response is mostly reasonable and factually correct but may include minor mistakes or unsupported assumptions.
>
> 4 points: The response is accurate and logically sound, with only negligible or borderline issues.
>
> 5 points: The response is fully accurate and logically consistent. It reflects a correct and well-reasoned understanding of the user's input.
>
> Below are the transcription of user's instruction and models' response:
> ### [Instruction]
> {}
>
> ### [Response]
> {}
>
> After evaluating, please output the score only without anything else. You don't need to provide any explanations.

### E.3 PROMPTS FOR GEMINI-2.5-PRO TO GENERATE ANNOTATIONS OF CONTEXT-GROUNDED VQA

You are given a video of a task execution, represented as a sequence of frames. Your task is to analyze each frame independently and answer the question: '{}' You must choose exactly one answer for each frame from the following options: {}. If consecutive frames have the same result, merge them into a single range. Format your output as: Frames <start_index>-<end_index>: <chosen option>

### E.4 PROMPTS FOR GEMINI-2.5-PRO TO EVALUATE THE ACCURACY OF RESPONSE ON CONTEXT-GROUNDED VQA

You are given:
1. A task description outlining the sequential actions the robot must perform.
2. A video showing the robot successfully executing this task.
3. A specific frame captured during the task execution.
4. A question about this frame, whose correct answer may vary depending on the
task's progress.
5. The robot's answer to the question.
Your job is to carefully evaluate whether the robot's answer is correct, considering the task description, the full video context, and the given frame. Respond with only one word: "Right." if the answer is correct, or "Wrong." if the answer is incorrect. Do not provide explanations.
[INPUT]
task description:{}
question:{}
answer:{}

### E.5 PROMPTS FOR GEMINI-2.5-PRO TO GENERATE ANNOTATIONS OF DEFECTIVE INSTRUCTION REJECTION

Look at the input image. Your task is to propose a non-executable robotic arm action command within the current scene. There are four categories of non-executable instructions:
1. visual: A task referencing visual characteristics (color, texture, surface, shape) that do not exist in the scene.
2. semantic: A task referencing an object that does not exist in the scene.
3. motion: A task involving a motion that is impossible due to the structure or
kinematic limitations of the robotic arm.
4. out-of-context: A completely unrelated, illogical, or nonsensical command.

You MUST generate an instruction of the following category: {}.
Then, provide a response from the assistant's perspective explaining why the command cannot be executed.
Output ONLY a valid Python dictionary in JSON format WITHOUT code block markers or extra text:
{'Category': '{}', 'Instruction': xxx, 'Response': xxx}

## F FURTHER DISCUSSION OF SA-MoE

We select the speech expert and the action expert as the foundation of ELLSA primarily because individual experts are readily accessible. However, this is not the only viable approach. SA-MoE provides a generalizable framework capable of integrating multimodal processing from any set of experts. Here, we would like to highlight an alternative solution that we consider more elegant and efficient: a universal backbone capable of processing all input modalities as the "brain", supported

by specialized encoders such as the speech encoder as the "ear", and the vision encoder as the "eye"; text expert as the "mouth", and the action expert as the "hand". Such a "brain-hand-mouth" architecture more closely mirrors the natural information flow of humans.

Several studies have explored multimodal processing with MoE to enhance performance (Lin et al., 2024; Li et al., 2025b; He et al., 2025) (Diao et al., 2025; Luo et al., 2025). However, SA-MoE differs from these works not only in the expert interaction mechanism but also the scope. Prior work focuses primarily on vision understanding, using MoE to alleviate domain conflicts, whereas SA-MoE integrates pretrained experts across diverse modalities to address modality interference. Moreover, from the perspective of training, existing approaches primarily serve as superior pretraining architectures, while SA-MoE is a high-performance and data-efficient architecture for post-training.

We compare SA-MoE against a single dense model. SA-MoE is finetuned with LoRA for 500 steps with all tasks, while the dense model is fully finetuned for 3k steps with only basic tasks, speech interaction and robot manipulation. Results in Table 7 show that SA-MoE significantly outperforms the dense model. Although the dense model can partially inherit capabilities from its initialized components, it struggles to learn effectively from unfamiliar modalities given the limited scale of our training data, far smaller than typical pretraining corpora. Moreover, SA-MoE surpasses the dense model in modalities corresponding to the initialized model, demonstrating that SA-MoE effectively addresses the modality interference problem that hampers dense models. We further compare SA-MoE with its individual experts, as shown in Table 8. Results indicate that SA-MoE largely preserves expert-level performance, with a relative decline of 10.3% for the speech expert and 6.4% for the action expert. We assume that the greater drop in the speech expert's performance may be attributed to sequence length differences: a single image typically generates around 300 tokens, while a 10-second speech sample yields only about 50 tokens, making alignment more challenging.

In summary, SA-MoE proves to be an efficient and effective strategy for integrating experts. It not only achieves robust modality integration and mitigates modality interference, but also leverages and retains much of the pretrained capability of each individual expert.

Table 7: Comparison between SA-MoE and using one dense model

| Model | Llama Q. | Web Q. | TriviaQA | AlpacaEval |
|---|---|---|---|---|
| Dense (from speech expert) | 62.7 | 23.6 | 29.7 | 2.12 |
| Dense (from action expert) | 32.7 | 3.9 | 9.1 | 1.25 |
| SA-MoE | 74.7 | 39.5 | 45.2 | 3.09 |

(a) Speech interaction S2T performance.

| Model | SPATIAL | OBJECT | GOAL | LONG |
|---|---|---|---|---|
| Dense (from speech expert) | 0.2% | 3.2% | 5.2% | 0.0% |
| Dense (from action expert) | 76.8% | 76.4% | 67.8% | 60.6% |
| SA-MoE | 90.8% | 95.8% | 86.4% | 84.4% |

(b) Robot manipulation performance.

Table 8: Comparison between ELLSA and individual experts

| Model | Llama Q. | Web Q. | TriviaQA | AlpacaEval |
|---|---|---|---|---|
| Speech expert | 77.7 | 44.1 | 55.9 | 3.35 |
| SA-MoE | 74.7 | 39.5 | 45.2 | 3.09 |

(a) Speech interaction S2T performance.

| Model | SPATIAL | OBJECT | GOAL | LONG |
|---|---|---|---|---|
| Action expert | 95.4% | 98.8% | 93.6% | 94.0% |
| SA-MoE | 90.8% | 95.8% | 86.4% | 84.4% |

(b) Robot manipulation performance.

# G  ABLATION STUDIES

## G.1  TIME BLOCK DURATION

To examine the effect of the time block duration, we conducted an ablation study by implementing a version of ELLSA that operates at 0.48 seconds. (We use 0.48 s instead of 0.5 s because the speech encoder runs at 25 Hz and cannot support a 0.5 s interval.) The results are shown in Table 9. We first trained individual experts compatible with this finer-grained block: a speech expert that generates 4 text tokens every 0.48 s and an action expert that produces 5 action frames per block. The results show that the two speech experts achieve comparable performance, whereas the action expert exhibits a notable degradation. This drop is likely due to the shorter action sequences, which can reduce the temporal coherence of the generated actions. As a consequence, the performance of SA-MoE at the 0.48 s timescale is largely influenced by the limitations of its action expert.

For latency, we report the average per–time-block latency measured on an A100 GPU. Both configurations complete inference within their respective time blocks, confirming that the full-duplex interaction loop remains feasible. Moreover, using a smaller time block naturally reduces interaction latency. We also expect that latency can be further improved through optimized inference frameworks such as vLLM and by deploying the model on more powerful GPUs.

Table 9: Ablation study on time block duration.

| Time block | Llama Q. | Web Q. | TriviaQA | AlpacaEval |
|---|---|---|---|---|
| 1s | 77.7 | 44.1 | 55.9 | 3.35 |
| 0.48s | 78.5 | 44.8 | 55.9 | 3.68 |

(a) Speech expert performance.

| Time block | SPATIAL | OBJECT | GOAL | LONG |
|---|---|---|---|---|
| 1s | 95.4% | 98.8% | 93.6% | 94.0% |
| 0.48s | 91.0% | 92.4% | 84.2% | 81.0% |

(b) Action expert performance.

| Time block | Speech Interaction | | | | Robot manipulation | | | |
|---|---|---|---|---|---|---|---|---|
| | Llama Q. | Web Q. | TriviaQA | AlpacaE. | SPATIAL | OBJECT | GOAL | LONG |
| 1s | 74.7 | 39.5 | 45.2 | 3.09 | 90.8% | 95.8% | 86.4% | 84.4% |
| 0.48s | 71.7 | 38.4 | 44.7 | 2.95 | 81.0% | 85.0% | 73.4% | 71.6% |

(c) SA-MoE performance.

| Time block | Speech-to-speech | Speech-to-action |
|---|---|---|
| 1s | 854ms | 786ms |
| 0.48s | 455ms | 428ms |

(d) Per-time-block latency.

## G.2  NUMBER OF EXPERTS

We conducted ablation studies on the number of experts, with results summarized in Table 11. We compare the 2-expert SA-MoE with a 3-expert variant, implemented by splitting either (1) vision and action into separate experts or (2) speech and text into separate experts. The separated experts are initialized from the same model. The results show that the 2-expert ELLSA performs on par with both 3-expert configurations. Given this similar performance and the efficiency of training fewer experts, we adopt the 2-expert design for ELLSA.

Table 10: Ablation study on the number of experts.

| Number of experts | Llama Q. | Web Q. | TriviaQA | AlpacaEval |
|---|---|---|---|---|
| 2 (Speech, Action) | 74.7 | 39.5 | 45.2 | 3.09 |
| 3 (Speech, Vision, Action) | 70.3 | 35.9 | 45.9 | 3.08 |
| 3 (Speech, Text, Action) | 73.0 | 34.7 | 45.0 | 2.93 |

(a) Speech interaction S2T performance.

| Number of experts | SPATIAL | OBJECT | GOAL | LONG |
|---|---|---|---|---|
| 2 (Speech, Action) | 90.8% | 95.8% | 86.4% | 84.4% |
| 3 (Speech, Vision, Action) | 89.8% | 96.2% | 85.8% | 84.4% |
| 3 (Speech, Text, Action) | 89.0% | 97.4% | 87.2% | 87.8% |

(b) Robot manipulation performance.

## G.3 SPEECH ENCODER

To investigate the impact of stronger components, we built an enhanced version of ELLSA by replacing its speech encoder with a stronger model, SPEAR (Yang et al., 2025), a zipformer encoder trained on substantially larger datasets and optimized with more specialized training strategies. With this upgraded encoder, ELLSA not only exhibits improved baseline capability, but also importantly, the performance gap in the speaking-while-acting scenario is markedly reduced, for example from 13.3% to 2.9% on LIBERO LONG, and from 17.0% to 7.0% on Web Questions. The results demonstrate that the performance degradation of speaking-while-acting arises primarily from the model capacity limit rather than from the architecture itself.

Table 11: Ablation study on speech encoder.

| Speech Encoder | Speaking Alone | | | | Speaking-while-acting | | | |
|---|---|---|---|---|---|---|---|---|
| | Llama Q. | Web Q. | TriviaQA | AlpacaE. | Llama Q. | Web Q. | TriviaQA | AlpacaE. |
| mamba | 74.7 | 39.5 | 45.2 | 3.09 | 68.9 (-7.8%) | 32.8 (-17.0%) | 35.1 (-22.3%) | 2.66 (-13.9%) |
| SPEAR | 76.7 | 48.7 | 61.5 | 3.81 | 74.8 (-2.5%) | 45.3 (-7.0%) | 53.2 (-13.5%) | 3.67 (-3.7%) |

(a) Speech interaction S2T performance.

| Speech Encoder | Acting Alone | | | | Speaking-while-acting | | | |
|---|---|---|---|---|---|---|---|---|
| | SPATIAL | OBJECT | GOAL | LONG | SPATIAL | OBJECT | GOAL | LONG |
| mamba | 90.8% | 95.8% | 86.4% | 84.4% | 93.3% (+2.8%) | 96.6% (+0.8%) | 86.1% (-0.3%) | 73.2% (-13.3%) |
| SPEAR | 91.2% | 96.6% | 90.4% | 88.8% | 90.9% (-0.3%) | 96.6% (-0.0%) | 89.7% (-0.8%) | 86.2% (-2.9%) |

(b) Robot manipulation performance.

## H RESULTS ON CALVIN BENCHMARK

To validate the effectiveness of SA-MoE across diverse benchmarks, we also develop an ELLSA variant for the CALVIN benchmark (Mees et al., 2022). This variant is built upon the SA-MoE framework trained on all basic tasks, combining the SPEAR-based speech expert with a CALVIN-specific UniVLA as the action expert. For CALVIN, we follow the standard ABCD→D protocol, training on four simulated environments (A, B, C, D) and testing on environment D. In CALVIN, the agent must complete a sequence of 5 tasks, where failure at any step terminates the remainder of the sequence. Consistent with prior work, we report both success rate at each task index, and the average number of consecutively completed tasks. The results show that our SA-MoE framework extends naturally to other VLA benchmarks by plugging in task-specific finetuned action experts. Moreover, the CALVIN variant achieves performance comparable to (and in some cases exceeding) modality-specific baselines. These findings further support our central claim that SA-MoE is an efficient, scalable, end-to-end full-duplex framework capable of handling concurrent multimodal input and output streams across diverse embodied learning settings.

Table 12: ELLSA performance on CALVIN benchmark.

| Model | Llama Q. | Web Q. | TriviaQA | AlpacaEval |
|---|---|---|---|---|
| Moshi(Défossez et al., 2024) | 60.8 | 23.4 | 25.6 | 1.84 |
| Freeze-Omni(Wang et al., 2025a) | 74.2 | 40.8 | 45.1 | 3.90 |
| ELLSA | 78.7 | 50.1 | 65.1 | 4.00 |

(a) Speech interaction S2T performance.

| Model | Tasks Completed in a Row | | | | | Avg. Len ↑ |
|---|---|---|---|---|---|---|
| | 1 | 2 | 3 | 4 | 5 | |
| MCIL(Lynch & Sermanet, 2020) | 0.373 | 0.027 | 0.002 | 0.000 | 0.000 | 0.40 |
| RT-1(Brohan et al., 2022) | 0.844 | 0.617 | 0.438 | 0.323 | 0.227 | 2.45 |
| Robo-Flamingo(Li et al., 2024b) | 0.964 | 0.896 | 0.824 | 0.740 | 0.660 | 4.09 |
| GR-1(Wu et al., 2024) | 0.949 | 0.896 | 0.844 | 0.789 | 0.731 | 4.21 |
| UP-VLA(Zhang et al., 2025) | 0.962 | 0.921 | 0.879 | 0.842 | 0.812 | 4.42 |
| RoboVLMs(Li et al., 2024a) | 0.967 | 0.930 | 0.899 | 0.865 | 0.826 | 4.49 |
| ELLSA | 0.967 | 0.928 | 0.889 | 0.849 | 0.793 | 4.43 |

(b) Robot manipulation performance.

# I FURTHER DETAILED RESULTS

For context-grounded VQA, Gemini may occasionally misjudge accuracy because it processes videos at only 1 fps and therefore cannot take all frames into account, potentially missing essential contextual information.

Table 13: Detailed results when dealing with different types of speech input during action execution (i.e., the speaking-while-acting task).

| Dataset | General question | Interruptive command | Silence |
|---|---|---|---|
| SPATIAL | 100% | 96% | 100% |
| OBJECT | 100% | 96% | 100% |
| GOAL | 100% | 89% | 100% |
| LONG | 100% | 96% | 100% |

Table 14: Detailed results for both speaking and acting on speaking-while-acting. The results are robot manipulation success rate %, S2T speech interaction performance and S2S speech interaction performance, respectively.

| | Llama Q. | Web Q. | TriviaQA | AlpacaEval |
|---|---|---|---|---|
| **SPATIAL** | 94.6%/71.0/65.0 | 92.2%/33.8/28.7 | 93.8%/34.5/30.2 | 92.4%/2.65/2.19 |
| **OBJECT** | 96.4%/69.3/65.7 | 97.2%/34.0/28.7 | 96.6%/36.0/31.8 | 96.2%/2.63/2.10 |
| **GOAL** | 84.8%/69.3/62.7 | 86.4%/32.2/26.9 | 90.2%/36.7/31.7 | 82.8%/2.69/2.06 |
| **LONG** | 74.2%/66.0/57.3 | 73.6%/31.1/26.1 | 73.6%/33.3/29.2 | 71.2%/2.68/2.14 |

Table 15: Detailed results on context-grounded VQA task

| Question Num | 1 | 2 | 3 | 4 | 5 | 6 | 7 | 8 | 9 | 10 | 11 | 12 |
|---|---|---|---|---|---|---|---|---|---|---|---|---|
| **Manual Acc.%** | 100 | 70 | 100 | 100 | 90 | 60 | 100 | 70 | 60 | 100 | 60 | 90 |
| **Gemini Acc.%** | 100 | 80 | 70 | 100 | 100 | 100 | 90 | 70 | 40 | 70 | 90 | 100 |

## J    LIMITATIONS

ELLSA still faces several limitations. First, although ELLSA successfully predicts duplex dynamics such as dialogue/action turn-taking and action barge-ins, it currently handles only a limited range of scenarios. Many aspects of natural communication, such as user and assistant backchanneling, remain unaddressed. Expanding support for these dynamics will be crucial for achieving more human-like interactions. Second, although ELLSA has shown promising results in handling full-duplex multimodal joint perception and concurrent generation within simulated environments, ELLSA has yet to be validated in real-world settings. Future work will focus on real-world deployment, and we believe our framework can be effectively adapted through targeted finetuning.

## K    THE USE OF LARGE LANGUAGE MODELS (LLMS)

For paper-writing, we use GPT-5 and Gemini-2.5-Pro to polish writing and find related work. We also use Doubao to generate the logo of ELLSA.

For experiments, we use LLM to generate training data (Llama-3-Instruct and Gemini-2.5-Pro) and evaluate results for AlpacaEval (GPT-4.1-2025-04-14) and context-grounded VQA (Gemini-2.5-Pro), with prompts provided in Appendix E.

