# OpenReview forum: "End-to-end Listen, Look, Speak and Act"
_ICLR.cc/2026/Conference — ICLR 2026 Poster_

### Official Review · Reviewer_smUE · 2025-10-30

**Soundness:** 3
**Presentation:** 3
**Contribution:** 2
**Rating:** 4
**Confidence:** 3

**Summary:**

This paper propose ELLSA, a unified, full-duplex multimodal model capable of simultaneous perception and generation across speech, vision, text, and action modalities. Specifically, the model proposes SA-MoE that routes inputs to modality-specific experts and fuses them through shared attention. Multimodal streams are partitioned into 1-second segments to support full-duplex interaction. The experiments show that ELLSA achieves performance comparable to or surpassing modality-specific baselines on multiple benchmarks covering speech interaction, speech-conditioned robot manipulation, and multimodal full-duplex tasks.

**Strengths:**

- The paper makes the first attempt to explore end-to-end full-duplex multimodal interaction, pointing to a promising new direction.
- The paper conduct experiments on benmarks of various task type and demonstrate competitive performance.

**Weaknesses:**

- The novelty of SA-MoE is limited, as prior works such as EVE-v2 [1] and Mono-InternVL [2] already adopt modality-specific MoE structures.
- The ablation study is insufficient and could be expanded to better demonstrate the contributions of individual components.
- The discussion of computational efficiency and real-time constraints is limited and should be addressed more thoroughly.
- The experiments on VLA tasks are limited, lacking experiments on more benchmarks and real world experiments.

References
[1] Diao, Haiwen, et al. “EVE-v2: Improved baselines for encoder-free vision-language models.” arXiv preprint arXiv:2502.06788 (2025).
[2] Luo, Gen, et al. “Mono-InternVL: Pushing the boundaries of monolithic multimodal large language models with endogenous visual pre-training.” Proceedings of the Computer Vision and Pattern Recognition Conference, 2025.

**Questions:**

1. The paper would benefit from clarifying the novelty of SA-MoE compared to existing methods such as EVE-v2 or Mono-InternVL.
2. I encourage the authors to include additional ablation studies, for example:
    1. Evaluate the impact of time block duration on both performance and latency.
    2. Investigate whether increasing the number of experts—by splitting vision and action modalities into separate experts—would further improve overall performence , while the action model is currently responsible for processing both vision and action inputs.
3. Given the full-duplex streaming interaction setup, it is important to analyze the response latency of the entire system.
4. The VLA evaluation is currently limited to the LIBERO dataset, and the author should evaluate the model on additional benchmarks and real world experiments to better assess the model’s generalizability.

---

> ### Author Response · Authors · 2025-11-21
> **Response to Reviewer smUE**
>
> Thank you for the dedicated effort that you’ve put into reviewing. We sincerely appreciate your valuable feedback and constructive suggestions.
>
> **Weakness 1 & Question 1: The novelty of SA-MoE is limited, as prior works such as EVE-v2 and Mono-InternVL already adopt modality-specific MoE structures; The paper would benefit from clarifying the novelty of SA-MoE compared to existing methods.**
>
> Thank you for the question. We would first like to emphasize that the key novelty of **ELLSA** lies in being, to our knowledge, **the first end-to-end model that accepts both vision and speech inputs and produces text, speech, and action outputs in a full-duplex, streaming manner.** This capability is essential for building human-like, real-time multimodal interactive agents. With this context, we also clarify that ELLSA does not lack architectural or methodological novelty. While it uses modality-specific experts, similar to some prior works, our approach differs substantially in both scope and purpose compared to methods such as EVE-v2 and Mono-InternVL (see Appendix F).
>
> **EVE-v2** and **Mono-InternVL** operate solely on vision–text modalities and employ modality-specific MoE modules primarily as **pretraining architectures**. In contrast, **SA-MoE** is designed as a data-efficient, high-performance **post-training framework** that enables flexible integration of multiple pretrained experts. This distinction is particularly important in our setting, where strong experts exist for different modality subsets (e.g., a speech+text expert and a vision+action expert). SA-MoE provides a simple and scalable way to connect these heterogeneous experts into a unified multimodal interactive system.
>
> Furthermore, SA-MoE introduces an extended attention interaction mechanism that operates over **interleaved multimodal sequences** and enables **bidirectional cross-expert attention.** Unlike prior architectures, where, for example, a vision expert provides features to a text expert but does not itself attend back, SA-MoE enforces that every expert both contributes to and integrates information from others to produce coherent multimodal outputs. We have expanded the discussion of these related methods in Appendix F to highlight these conceptual and functional differences more clearly.
>
> ---
>
> **Weakness 2 & Question 2 (1): Evaluate the impact of time block duration on both performance and latency.**
>
> Thank you for this helpful question. We use a 1-second (sec.) block size as a practical compromise because the action expert operates at 10 FPS. This alignment provides stable action sequences while keeping the duplex loop responsive. To assess the impact of block size, we implemented a 0.48-second variant of ELLSA (0.48-sec. instead of 0.5-sec. because the speech encoder runs at 25 Hz). We retrained both experts under this finer-grained setting:
> - Speech expert: 4 text tokens per 0.48-sec.
> - Action expert: 5 action frames per 0.48-sec.
>
> The speech expert shows comparable performance across 1-sec. and 0.48-sec., confirming that speech can indeed run at a faster tempo.
>
> The action expert, however, exhibits a clear drop in performance at 0.48-sec., likely because shorter action sequences weaken temporal coherence. As a result, the overall SA-MoE performance at this timescale is bottlenecked by the action expert rather than speech processing.
>
> We measured average per-block latency on an A100 GPU. Both the 1-sec. and 0.48-sec. configurations complete inference well within their respective block windows, ensuring real-time full-duplex interaction remains feasible. Smaller blocks further reduce end-to-end latency, and we expect additional gains with optimised inference engines (e.g., vLLM) or more powerful GPUs.
>
> In short, although speech can operate at a finer temporal resolution, the action expert benefits from the longer 1-sec. block for stable control. This configuration matches the settings commonly adopted in VLA research and provides the best trade-off between responsiveness and action quality in the duplex setting.
>
> ### Ablation study on time block duration.
>
> **(a) Speech expert performance.**
>
> | Time block | Llama Q. | Web Q. | TriviaQA | AlpacaEval |
> | :--- | :--- | :--- | :--- | :--- |
> | 1s | 77.7 | 44.1 | 55.9 | 3.35 |
> | 0.48s | 78.5 | 44.8 | 55.9 | 3.68 |
>
> **(b) Action expert performance.**
>
> | Time block | SPATIAL | OBJECT | GOAL | LONG |
> | :--- | :--- | :--- | :--- | :--- |
> | 1s | 95.4% | 98.8% | 93.6% | 94.0% |
> | 0.48s | 91.0% | 92.4% | 84.2% | 81.0% |
>
> **(c) SA-MoE performance.**
>
> | Time block | Llama Q. | Web Q. | TriviaQA | AlpacaE. | SPATIAL | OBJECT | GOAL | LONG |
> | :--- | :--- | :--- | :--- | :--- | :--- | :--- | :--- | :--- |
> | 1s | 74.7 | 39.5 | 45.2 | 3.09 | 90.8% | 95.8% | 86.4% | 84.4% |
> | 0.48s | 71.7 | 38.4 | 44.7 | 2.95 | 81.0% | 85.0% | 73.4% | 71.6% |
>
> **(d) Per-time-block latency.**
>
> | Time block | Speech-to-speech | Speech-to-action |
> | :--- | :--- | :--- |
> | 1s | 854ms | 786ms |
> | 0.48s | 455ms | 428ms |

---

> ### Author Response · Authors · 2025-11-21
> **Response to Reviewer smUE (part2)**
>
> **Weakness 2 & Question 2 (2): Investigate whether increasing the number of experts—by splitting vision and action modalities into separate experts—would further improve overall performence , while the action model is currently responsible for processing both vision and action inputs.**
>
> Thank you for the insightful question. We choose to include two experts in ELLSA because the speech expert is trained on both speech+text, whereas the action expert is trained on both vision+action. Grouping speech with text and vision with action allows us to most effectively leverage the pretrained capabilities of each model, as described in Section 3.2.
>
> We also conducted ablations on the number of experts. Due to memory constraints, a full 4-expert MoE is not feasible, so we evaluated two 3-expert variants by splitting either (1) vision/action or (2) speech/text into separate experts (with split experts initialized from the same pretrained model). The results indicate that 3-expert versions do not bring stable and significant performance gain compared to the 2-expert SA-MoE. As discussed in Section 3.2, “We merge speech with text and vision with action to better leverage pretrained knowledge.” In our design, the speech expert is jointly trained on speech and text modalities, while the action expert is trained on both vision and action modalities. This alignment may explain why further subdividing the experts does not yield additional performance gains. Moreover, we also believe that if more powerful, specialized vision and action experts were available, splitting these modalities could further enhance overall performance. Given the additional efficiency of fewer experts, we adopt the 2-expert design. These discussions are now presented in Appendix G (Number of experts).
>
> ### Ablation study on the number of experts.
>
> **(a) Speech interaction S2T performance.**
>
> | Number of experts | Llama Q. | Web Q. | TriviaQA | AlpacaEval |
> | :--- | :--- | :--- | :--- | :--- |
> | 2 (Speech, Action) | 74.7 | 39.5 | 45.2 | 3.09 |
> | 3 (Speech, Vision, Action) | 70.3 | 35.9 | 45.9 | 3.08 |
> | 3 (Speech, Text, Action) | 73.0 | 34.7 | 45.0 | 2.93 |
>
> **(b) Robot manipulation performance.**
>
> | Number of experts | SPATIAL | OBJECT | GOAL | LONG |
> | :--- | :--- | :--- | :--- | :--- |
> | 2 (Speech, Action) | 90.8% | 95.8% | 86.4% | 84.4% |
> | 3 (Speech, Vision, Action) | 89.8% | 96.2% | 85.8% | 84.4% |
> | 3 (Speech, Text, Action) | 89.0% | 97.4% | 87.2% | 87.8% |
>
> ---
>
> **Weakness 3 & Question 3: The discussion of computational efficiency and real-time constraints is limited and should be addressed more thoroughly; Given the full-duplex streaming interaction setup, it is important to analyze the response latency of the entire system.**
>
> As shown in our response to Question 2(1), ELLSA achieves **854 ms** speech-to-speech latency and **786 ms** speech-to-action latency, both well within the 1-second streaming window. These measurements confirm that full-duplex streaming interaction is feasible with our current implementation. We also expect further latency reductions in real-world deployments by leveraging optimized inference frameworks (e.g., vLLM) and more powerful GPUs.
>
> ---
>
> **Weakness 4 & Question 4: The experiments on VLA tasks are limited, lacking experiments on more benchmarks and real world experiments. & The VLA evaluation is currently limited to the LIBERO dataset, and the author should evaluate the model on additional benchmarks and real world experiments to better assess the model’s generalizability.**
>
> Thank you for the valuable question. Since **LIBERO is the only action-related dataset used for training**, the current ELLSA model is not expected to perform competitively on other action benchmarks or real-world tasks due to the inherent training–test mismatch. For example, if a new benchmark includes actions such as closing a laptop, but no laptop-related demonstrations exist in the training set, the model would naturally struggle to interpret and execute such instructions. Evaluating ELLSA on additional datasets would therefore require training a new model, which would differ from the version reported in this paper.
>
> It is also important to clarify that our goal is **not** to build a fully task-level generalisable action agent. Instead, the primary contribution of this work is an efficient, scalable, end-to-end full-duplex framework capable of handling concurrent multimodal input–output streams. The LIBERO benchmark, which is designed to measure knowledge transfer in lifelong robot learning, already demonstrates that our framework can generate accurate and consistent action outputs in a realistic setting. With access to more diverse and comprehensive action datasets, we expect the framework to support models with significantly stronger generalization capabilities. We also plan to develop ELLSA variants tailored for additional simulated environments and real-world settings in future work.

---

> ### Author Response · Authors · 2025-11-27
> **Further Response to Reviewer smUE**
>
> Regarding Weakness 4, we would like to clarify that we have also developed an ELLSA variant for the **CALVIN benchmark** [1], a widely used evaluation suite in VLA research. This variant is built upon the SA-MoE framework, combining the SPEAR-based speech expert with a CALVIN-specific action expert trained on all basic tasks. For CALVIN, we follow the standard ABCD→D protocol, training on four simulated environments (A, B, C, D) and testing on environment D. The results are provided below.
> In CALVIN, the agent must complete a sequence of 5 tasks, where failure at any step terminates the remainder of the sequence. Consistent with prior work, we report both:
> 1. success rate at each task index, and
> 2. the average number of consecutively completed tasks.
>
> The results show that our SA-MoE framework **extends naturally to other VLA benchmarks** by plugging in task-specific finetuned action experts. Moreover, the CALVIN variant achieves performance **comparable to (and in some cases exceeding) modality-specific baselines**. These findings further support our central claim that SA-MoE is an **efficient, scalable, end-to-end full-duplex framework** capable of handling concurrent multimodal input and output streams across diverse embodied learning settings.
>
> [1] Mees, Oier, et al. "CALVIN: A benchmark for language-conditioned policy learning for long-horizon robot manipulation tasks." IEEE Robotics and Automation Letters 7.3 (2022): 7327-7334.
>
> ### ELLSA performance on CALVIN benchmark
>
> **(a) Speech interaction S2T performance**
>
> | Model | Llama Q. | Web Q. | TriviaQA | AlpacaEval |
> | :--- | :--- | :--- | :--- | :--- |
> | Moshi | 60.8 | 23.4 | 25.6 | 1.84 |
> | Freeze-Omni | 74.2 | 40.8 | 45.1 | 3.90 |
> | ELLSA | 78.7 | 50.1 | 65.1 | 4.00 |
>
> **(b) Robot manipulation performance**
>
> | Model | 1 | 2 | 3 | 4 | 5 | Avg. Len ↑ |
> | :--- | :--- | :--- | :--- | :--- | :--- | :--- |
> | MCIL | 0.373 | 0.027 | 0.002 | 0.000 | 0.000 | 0.40 |
> | RT-1 | 0.844 | 0.617 | 0.438 | 0.323 | 0.227 | 2.45 |
> | Robo-Flamingo | 0.964 | 0.896 | 0.824 | 0.740 | 0.660 | 4.09 |
> | GR-1 | 0.949 | 0.896 | 0.844 | 0.789 | 0.731 | 4.21 |
> | UP-VLA | 0.962 | 0.921 | 0.879 | 0.842 | 0.812 | 4.42 |
> | RoboVLMs | 0.967 | 0.930 | 0.899 | 0.865 | 0.826 | 4.49 |
> | ELLSA | 0.967 | 0.928 | 0.889 | 0.849 | 0.793 | 4.43 |

---

> ### Author Response · Authors · 2025-11-27
> **Request of Reviewer's attention and feedback**
>
> Dear Reviewer smUE,
>
> As a gentle reminder, it has been more than six days since we submitted our rebuttal, and we wanted to kindly check whether our responses have addressed your main concerns. If anything remains unclear or you would like further clarification or experiments, we would be very happy to provide them within the remaining discussion time.
>
> Thank you again for your valuable and constructive feedback, which has inspired further improvements to our paper.
>
> Best regards,
>
> The Authors

---

### Official Review · Reviewer_yfkh · 2025-10-30

**Soundness:** 3
**Presentation:** 3
**Contribution:** 3
**Rating:** 6
**Confidence:** 3

**Summary:**

This paper introduces ELLSA (End-to-end Listen, Look, Speak and Act), a novel end-to-end, full-duplex model. The motivation is to bridge the gap between disembodied conversational agents ("talkers") and non-conversant robotic agents ("doers") by creating a unified system that can simultaneously perceive multimodal inputs (vision, speech) and generate multimodal outputs (speech, actions). The core technical contribution is the Self-Attention Mixture-of-Experts (SA-MoE) architecture. This design routes different modalities to specialized, pre-trained expert models (a speech expert for speech/text and an action expert for vision/action) and integrates them through a shared self-attention mechanism. The authors shows ELLSA's capabilities on a wide range of tasks. It achieves competitive or superior performance on standard benchmarks for speech interaction and speech-conditioned robot manipulation.

**Strengths:**

- The paper addresses a fundamental and ambitious goal in AI: creating truly interactive embodied agents that can communicate and act in a fluid, human-like manner.

- The proposed SA-MoE architecture is a well-motivated to the problem of modality interference, a common challenge in multimodal model development. The ablation study comparing SA-MoE to a dense model (Table 7) provides compelling evidence of its superiority in performance.

- The paper shows its evaluation of the unique capabilities enabled by its full-duplex, multimodal design. The successful demonstration of action turn-taking, barge-ins, and speaking-while-acting (Tables 3 and Table 4) directly validates the central claims of the paper and showcases interaction patterns that are qualitatively more natural and complex than what previous systems could achieve.

**Weaknesses:**

- The results for "speaking-while-acting" (Table 4) show a noticeable drop in performance for both the manipulation and speech interaction tasks compared to their single-task counterparts (Tables 1 & 2). The authors note that the model "may be distracted," which is an honest assessment. This performance hit raises questions about the scalability of this approach to more complex, simultaneous tasks and the true cost of concurrent generation.

- The system relies on distinct modality-specific components and different backbones. It will be better with a more unified architecture which  could enable deeper cross-modal fusion and represent a more fundamental approach to generalized interactive AI.

**Questions:**

- Could you elaborate on the performance degradation observed during the speaking-while-acting task? Is this an issue of model capacity (i.e., would a larger backbone help?), an architectural limitation of SA-MoE where attention is being split, or a data-related artifact?

- The concept of full-duplex interaction is intrinsically tied to low latency.  Could you provide any measurements for key interaction loops, such as speech-to-action or speech-to-speech-response times?

---

> ### Author Response · Authors · 2025-11-21
> **Response to Reviewer yfkh**
>
> We thank the reviewer for the positive comments and would like to clarify and address each concern as follows:
>
> **Weakness 1 & Question 1: The results for "speaking-while-acting" (Table 4) show a noticeable drop in performance for both the manipulation and speech interaction tasks compared to their single-task counterparts (Tables 1 & 2). The authors note that the model "may be distracted," which is an honest assessment. This performance hit raises questions about the scalability of this approach to more complex, simultaneous tasks and the true cost of concurrent generation.; Could you elaborate on the performance degradation observed during the speaking-while-acting task? Is this an issue of model capacity (i.e., would a larger backbone help?), an architectural limitation of SA-MoE where attention is being split, or a data-related artifact?**
>
> Thank you for the insightful question. We first note that speaking-while-acting is **inherently more challenging than speaking or acting alone.** In this setting, the model must perform two tasks concurrently, which naturally introduces competition for attention and increases the likelihood of distraction, as discussed in Section 5.2.2. A performance drop relative to single-modality tasks is therefore expected.
>
> Second, we argue that this degradation arises primarily from **the model capacity limit** rather than from the architecture itself. To validate this point, we built an enhanced version of ELLSA by replacing its speech encoder with a stronger model, SPEAR [1], a Zipformer encoder trained on substantially larger datasets and optimised with more specialised training strategies. With this upgraded encoder, ELLSA not only exhibits improved baseline capability, but also importantly, the performance gap in the speaking-while-acting scenario is markedly reduced. For example, from 13.3% to 2.9% on "LIBERO LONG" and from 17.0% to 7.0% on "web_questions".
>
> [1] Yang, Xiaoyu, et al. "SPEAR: A Unified SSL Framework for Learning Speech and Audio Representations." arXiv preprint arXiv:2510.25955 (2025). https://huggingface.co/collections/marcoyang/spear-encoders
>
> ### Ablation study on speech encoder.
>
> **(a) Speech interaction S2T performance.**
>
> | Speech Encoder |  Speaking alone | Llama Q. | Web Q. | TriviaQA  | AlpacaE. | Speaking-while-acting | Llama Q. | Web Q. | TriviaQA  | AlpacaE. |
> | :--- | :--- | :--- | :--- | :--- | :--- | :--- | :--- | :--- | :--- | :--- |
> | mamba | | 74.7 | 39.5 | 45.2 | 3.09 | | 68.9 (-7.8%) | 32.8 (-17.0%) | 35.1 (-22.3%) | 2.66 (-13.9%) |
> | SPEAR | | 76.7 | 48.7 | 61.5 | 3.81 | | 74.8 (-2.5%) | 45.3 (-7.0%) | 53.2 (-13.5%) | 3.67 (-3.7%) |
>
> **(b) Robot manipulation performance.**
>
> | Speech Encoder |  Acting alone | SPATIAL | OBJECT | GOAL | LONG | Speaking-while-acting | SPATIAL | OBJECT  | GOAL  | LONG |
> | :--- | :--- | :--- | :--- | :--- | :--- | :--- | :--- | :--- | :--- | :--- |
> | mamba | | 90.8% | 95.8% | 86.4% | 84.4% | | 93.3% (+2.8%) | 96.6% (+0.8%) | 86.1% (-0.3%) | 73.2% (-13.3%) |
> | SPEAR | | 91.2% | 96.6% | 90.4% | 88.8% | | 90.9% (-0.3%) | 96.6% (-0.0%) | 89.7% (-0.8%) | 86.2% (-2.9%) |

---

> ### Author Response · Authors · 2025-11-21
> **Response to Reviewer yfkh (part2)**
>
> **Weakness 2: The system relies on distinct modality-specific components and different backbones. It will be better with a more unified architecture which could enable deeper cross-modal fusion and represent a more fundamental approach to generalized interactive AI.**
>
> Thank you for this insightful comment. We fully agree that a more unified architecture is an appealing long-term direction, and that deeper cross-modal fusion could, in principle, yield a more fundamental solution to generalised interactive AI. However, building a single end-to-end model that jointly handles speech, vision, action, and text poses substantial practical challenges, including the need for extremely large multimodal datasets and carefully designed training strategies to handle the highly imbalanced distribution of data across modalities.
>
> In practice, many existing pretrained models cover only a subset of these modalities (e.g., speech+text or vision+action). Reusing these modality-specialised experts allows us to retain their strengths and avoid retraining large components from scratch, leading to a far more data- and compute-efficient solution. Our SA-MoE framework is designed precisely for this: it provides a flexible mechanism for integrating arbitrary pretrained experts while still allowing rich cross-modal interactions.
>
> Although a monolithic model may seem conceptually simpler, a modular design offers complementary advantages. It preserves interpretability, makes it easier to analyse or improve individual components, and ensures that advancements in any modality expert directly benefit the overall system. As discussed in Appendix F, our current architecture is not intended to be the only or universally optimal MoE design. Rather, SA-MoE offers a generalisable template that can support multiple instantiations.
>
> One particularly elegant instantiation (**aligned with your suggestion**) is a universal backbone acting as the “brain,” coupled with specialised modality encoders as the “ear” (speech input), “eye” (visual input), “mouth” (text and speech outputs), and “hand” (action). This type of “brain–ear–eye–mouth–hand” architecture mirrors natural human sensory and motor processing and is a promising direction building **human-like real-time multimodal interactive agents**.
>
> ---
>
> **Question 2: The concept of full-duplex interaction is intrinsically tied to low latency. Could you provide any measurements for key interaction loops, such as speech-to-action or speech-to-speech-response times?**
>
> Thank you for the valuable question. We evaluate the average per–time-block latency on an A100 GPU and find that ELLSA achieves speech-to-speech and speech-to-action latencies of **854 ms** and **786 ms**, respectively, both comfortably within the 1-second time block. These results confirm that full-duplex streaming interaction is fully achievable. In real-world deployments, we anticipate even lower latency by using optimised inference frameworks such as vLLM and by running the model on more capable GPUs.

---

> ### Author Response · Authors · 2025-11-27
> **Request of Reviewer's attention and feedback**
>
> Dear Reviewer yfkh,
>
> As a gentle reminder, it has been more than six days since we submitted our rebuttal, and we wanted to kindly check whether our responses have addressed your main concerns. If anything remains unclear or you would like further clarification or experiments, we would be very happy to provide them within the remaining discussion time.
>
> Thank you again for your valuable and constructive feedback, which has inspired further improvements to our paper.
>
> Best regards,
>
> The Authors

---

### Official Review · Reviewer_9jnm · 2025-11-01

**Soundness:** 3
**Presentation:** 3
**Contribution:** 3
**Rating:** 8
**Confidence:** 4

**Summary:**

This paper introduces ELLSA, an end-to-end, full-duplex model that seamlessly integrates listening, looking, speaking, and acting. Its novel core is the SA-MoE architecture, which routes modality-specific experts through a shared attention backbone to mitigate interference and enhance fusion. Evaluated on speech QA, robot manipulation, and VQA, ELLSA not only achieves strong performance but also demonstrates unique capabilities like speaking-while-acting and action barge-in.

**Strengths:**

1. This work presents a novel and practically significant real-time, full-duplex system that seamlessly integrates the speech, vision, and action modalities.
2. The proposed SA-MoE architecture offers a clean and modular solution to the challenge of modality interference by routing modalities to specialized components.
3. The system demonstrates remarkable versatility by tackling a diverse range of tasks from dialogue to manipulation.

**Weaknesses:**

1. The SA-MoE architecture is primarily evaluated empirically; new conceptual insight and theoretical analysis of its properties are lacking.
2. The study does not investigate the model's ability to generalize to unseen tasks or domains, no evidence of zero-shot or cross-task generalization is fully shown.
3. The broader implications for AGI remain speculative, as the model's capabilities are confined to the specific multimodal tasks presented. This seems to be an overstated narrative about AGI.

**Questions:**

1. Please explain the specific mechanism through which SA-MoE's cross-modal attention prevents interference. Is there quantitative evidence to support this?
2. What is the impact of the 1-second block size on real-time performance and the quality of the model's responses in a duplex setting?
3. Has the model been evaluated on unseen task or modality combinations to rigorously test its generalization capabilities?

---

> ### Author Response · Authors · 2025-11-21
> **Response to Reviewer 9jnm**
>
> We sincerely appreciate the high score you have given to our work. We hope the following content will further address and alleviate any remaining concerns you may have.
>
> **Weakness 1: The SA-MoE architecture is primarily evaluated empirically; new conceptual insight and theoretical analysis of its properties are lacking.**
>
> Thank you for the valuable question and for noting that "*SA-MoE offers a clean and modular solution to modality interference by routing modalities to specialized components.*" Below, we elaborate on the conceptual motivation and design insights behind SA-MoE.
>
> When building a model that jointly supports vision, speech, action, and text, we observed that strong pretrained models already exist for different modality subsets, for example, our speech expert (speech+text) and our action expert (vision+action). Given the limited availability of large-scale multimodal data, leveraging these pretrained experts is more feasible than training a unified model from scratch. This raises a central design question: **how can we integrate multiple pretrained experts into a single model while preserving their capabilities and enabling efficient cross-expert information exchange?**
>
> Our insight is grounded in how Transformers use **attention**. At inference time, a Transformer makes predictions by attending to its KV cache, which stores distilled high-level representations produced at previous steps. The attention module does not require these KV entries to originate from the same model, only that they match dimensionality and extract semantics. In other words, attention treats KV embeddings as generic information carriers. This suggests a natural integration strategy:
> - use **modality-specific pretrained experts** to extract representations for each modality, and
> - let **attention** serve as the unifying mechanism for fusing these representations.
>
> This approach avoids forcing a single backbone to learn both modality-specific feature extraction and multimodal integration from scratch when learning a new modality. Instead, SA-MoE allows each expert to focus solely on integration, while pretrained experts provide clean and meaningful representations. This significantly reduces learning complexity and makes the architecture scalable to additional modalities.
>
> Finally, we agree that there is ample room for further architectural advances that deepen cross-modal interaction. SA-MoE is intentionally simple, but it provides a principled and effective foundation for integrating pretrained experts through attention.
>
> ---
>
> **Weakness 2 & Question 3: Has the model been evaluated on unseen task or modality combinations?**
>
> Thank you for the question. Due to the limited scale of available training data, ELLSA does not yet exhibit strong zero-shot or cross-task generalisation. It does, however, generalize in several practical settings, for instance, following spoken instructions from unseen speakers, adapting to changes in scene layout, and answering unseen speech queries using pretrained knowledge.
>
> Our main contribution is introducing a data-efficient framework for full-duplex multimodal integration. ELLSA is the first end-to-end model that can simultaneously perceive and generate across vision, speech, text, and action, enabling behaviors such as speaking-while-acting, dialogue–action turn-taking, context-grounded VQA, and action barge-ins. We expect that with more diverse and larger-scale training data, the SA-MoE framework will support substantially stronger generalization in future full-duplex multimodal models.
>
> ### ELLSA's performance with action instructions from an unseen speaker (10 trails per instruction)
>
> | SPATIAL | OBJECT | GOAL | LONG |
> | :--- | :--- | :--- | :--- |
> | 85.0% | 96.0% | 85.0% | 90.0% |
>
> ---
>
> **Weakness 3: The broader implications for AGI remain speculative, as the model's capabilities are confined to the specific multimodal tasks presented. This seems to be an overstated narrative about AGI.**
>
> Thank you for the comment. We fully agree that our work does not achieve AGI, nor do we intend to claim it. Our goal is much narrower: to explore a capability we view as foundational for building **human-like real-time multimodal interactive agents, namely, full-duplex multimodal interaction.**
>
> Humans naturally listen, see, speak, and act simultaneously, adapting to turn-taking and interruptions in real time. Prior systems typically handle these abilities separately. In contrast, ELLSA is, to our knowledge, the first end-to-end full-duplex model that can jointly perceive and generate vision, speech, text, and action streams within a unified architecture.
>
> We acknowledge that ELLSA’s abilities remain limited, but it demonstrates several previously unattainable interactive behaviors (such as speaking-while-acting, dialogue–action turn-taking and action barge-ins) that we believe mark meaningful early steps in this direction. Following the reviewer’s suggestion, we will revise the paper accordingly.

---

> ### Author Response · Authors · 2025-11-21
> **Response to Reviewer 9jnm (part2)**
>
> **Question 1: Please explain the specific mechanism through which SA-MoE's cross-modal attention prevents interference. Is there quantitative evidence to support this?**
>
> Thank you for the question. Quantitative evidence supporting SA-MoE’s ability to reduce modality interference is provided in **Table 7** (also discussed in Section 5 and Appendix F). Dense baselines must jointly learn feature extraction and cross-modal integration from limited multimodal data. This often causes modalities to interfere with one another and leads to degradation of previously learned skills.
> SA-MoE avoids this issue by **routing each modality to its own pretrained expert**, which preserves modality-specific competence, and **using cross-modal attention only for integration**. This prevents experts from being forced to adapt to unfamiliar modalities. Empirically, SA-MoE achieves **substantially higher accuracy** than all dense baselines, providing quantitative evidence that its attention-based fusion mechanism effectively mitigates interference.
>
> ### Comparison between SA-MoE and using one dense model
>
> **(a) Speech interaction S2T performance**
>
> | Model | Llama Q. | Web Q. | TriviaQA | AlpacaEval |
> | :--- | :--- | :--- | :--- | :--- |
> | Dense (from speech expert) | 62.7 | 23.6 | 29.7 | 2.12 |
> | Dense (from action expert) | 32.7 | 3.9 | 9.1 | 1.25 |
> | SA-MoE | 74.7 | 39.5 | 45.2 | 3.09 |
>
> **(b) Robot manipulation performance**
>
> | Model | SPATIAL | OBJECT | GOAL | LONG |
> | :--- | :--- | :--- | :--- | :--- |
> | Dense (from speech expert) | 0.2% | 3.2% | 5.2% | 0.0% |
> | Dense (from action expert) | 76.8% | 76.4% | 67.8% | 60.6% |
> | SA-MoE | 90.8% | 95.8% | 86.4% | 84.4% |
>
> ---
>
> **Question 2: What is the impact of the 1-second block size on real-time performance and the quality of the model's responses in a duplex setting?**
>
> Thank you for this helpful question. We use a 1-second (sec.) block size as a practical compromise because the action expert operates at 10 FPS. This alignment provides stable action sequences while keeping the duplex loop responsive. To assess the impact of block size, we implemented a 0.48-second variant of ELLSA (0.48-sec. instead of 0.5-sec. because the speech encoder runs at 25 Hz). We retrained both experts under this finer-grained setting:
> - Speech expert: 4 text tokens per 0.48-sec.
> - Action expert: 5 action frames per 0.48-sec.
>
> The speech expert shows comparable performance across 1-sec. and 0.48-sec., confirming that speech can indeed run at a faster tempo.
>
> The action expert, however, exhibits a clear drop in performance at 0.48-sec., likely because shorter action sequences weaken temporal coherence. As a result, the overall SA-MoE performance at this timescale is bottlenecked by the action expert rather than speech processing.
>
> We measured average per-block latency on an A100 GPU. Both the 1-sec. and 0.48-sec. configurations complete inference well within their respective block windows, ensuring real-time full-duplex interaction remains feasible. Smaller blocks further reduce end-to-end latency, and we expect additional gains with optimized inference engines (e.g., vLLM) or more powerful GPUs.
>
> In short, although speech can operate at a finer temporal resolution, the action expert benefits from the longer 1-sec. block for stable control. This configuration matches the settings commonly adopted in VLA research and provides the best trade-off between responsiveness and action quality in the duplex setting.
>
> ### Ablation study on time block duration.
>
> **(a) Speech expert performance.**
>
> | Time block | Llama Q. | Web Q. | TriviaQA | AlpacaEval |
> | :--- | :--- | :--- | :--- | :--- |
> | 1s | 77.7 | 44.1 | 55.9 | 3.35 |
> | 0.48s | 78.5 | 44.8 | 55.9 | 3.68 |
>
> **(b) Action expert performance.**
>
> | Time block | SPATIAL | OBJECT | GOAL | LONG |
> | :--- | :--- | :--- | :--- | :--- |
> | 1s | 95.4% | 98.8% | 93.6% | 94.0% |
> | 0.48s | 91.0% | 92.4% | 84.2% | 81.0% |
>
> **(c) SA-MoE performance.**
>
> | Time block | Llama Q. | Web Q. | TriviaQA | AlpacaE. | SPATIAL | OBJECT | GOAL | LONG |
> | :--- | :--- | :--- | :--- | :--- | :--- | :--- | :--- | :--- |
> | 1s | 74.7 | 39.5 | 45.2 | 3.09 | 90.8% | 95.8% | 86.4% | 84.4% |
> | 0.48s | 71.7 | 38.4 | 44.7 | 2.95 | 81.0% | 85.0% | 73.4% | 71.6% |
>
> **(d) Per-time-block latency.**
>
> | Time block | Speech-to-speech | Speech-to-action |
> | :--- | :--- | :--- |
> | 1s | 854ms | 786ms |
> | 0.48s | 455ms | 428ms |

---

> ### Author Response · Authors · 2025-11-27
> **Request of Reviewer's attention and feedback**
>
> Dear Reviewer 9jnm,
>
> As a gentle reminder, it has been more than six days since we submitted our rebuttal, and we wanted to kindly check whether our responses have addressed your main concerns. If anything remains unclear or you would like further clarification or experiments, we would be very happy to provide them within the remaining discussion time.
>
> Thank you again for your valuable and constructive feedback, which has inspired further improvements to our paper.
>
> Best regards,
>
> The Authors

---

### Official Review · Reviewer_4T5n · 2025-11-03

**Soundness:** 2
**Presentation:** 2
**Contribution:** 2
**Rating:** 4
**Confidence:** 4

**Summary:**

This paper presents ELLSA (End-to-end Listen, Look, Speak and Act), claiming to be the first full-duplex, end-to-end model that simultaneously processes vision, speech, text, and action within a unified architecture. The core technical contribution is SA-MoE (Self-Attention Mixture-of-Experts), which routes modalities to specialized experts (Speech Expert and Action Expert) while enabling cross-modal fusion through shared attention. The model is trained in three stages and evaluated on speech interaction benchmarks and robotic manipulation tasks (LIBERO), demonstrating both competitive basic performance and novel full-duplex capabilities.

**Strengths:**

- This paper addresses full-duplex multimodal interaction for embodied AI is timely and important. The motivation for combining conversational abilities with physical action is well-articulated.

- SA-MoE is an elegant solution that leverages pretrained experts while mitigating modality interference.

- The introduction of context-grounded VQA during manipulation and systematic evaluation of full-duplex behaviors adds value beyond existing benchmarks.

**Weaknesses:**

- This paper omit discussion of prior work inlcuding:
   - Unified-IO 2 (Lu et al., 2023) which is also native support vision, laugnage, audio and action. The authors need to clearly differentiate from Unified-IO 2, likely arguing that their full-duplex streaming capability is the key difference.
  - Gato (Reed et al., 2022, DeepMind) which A generalist agent.
  - PaLM-E: Embodied multimodal model combining vision-language understanding and robotic manipulation.

- Limited Technical Novelty of SA-MoE: While effective, SA-MoE is essentially routing + shared attention between existing pretrained models.

- Speaking-while-acting shows notable performance drops (Table 4), especially on harder tasks (LIBERO LONG: 73.2% vs 84.4%)

**Questions:**

- Why specifically 2 experts rather than 4 (one per modality)? Have you experimented with different expert configurations? What is the computational overhead of SA-MoE vs a single model

- Can the model handle truly simultaneous inputs (e.g., receiving new speech while still processing previous speech and executing action)? The 1-second time block seems to discretize the interaction.

- What are the main challenges anticipated for real-world deployment? Have you done any preliminary real robot experiments?

---

> ### Author Response · Authors · 2025-11-21
> **Response to Reviewer 4T5n**
>
> We sincerely thank the reviewer for the dedicated effort and valuable suggestions. We hope the following responses will help address your concerns.
>
> **Weakness 1: This paper omit discussion of prior work**
>
> We thank the reviewer for highlighting these related works and have added them to the Related Work section. We further clarify the key distinctions below.
>
> The central differentiator of **ELLSA** is its ability to take vision and speech inputs and generate text, speech, and action outputs in an **end-to-end, full-duplex, streaming manner.** To our knowledge, no prior system supports continuous multimodal inputs and outputs simultaneously, which is essential for human-like real-time multimodal interactive agents.
>
> Regarding **Unified-IO 2**, although it covers all related modalities, it does not operate in a full-duplex manner and handles modalities largely in a task-separated fashion. For example, it can caption audio or predict actions independently, but cannot perform speech-conditioned action prediction or jointly reason over vision and speech. In contrast, ELLSA explicitly focuses on joint multimodal understanding, a crucial step toward building human-like intelligence. ELLSA can, for instance, analyze visual and speech inputs together to produce action outputs (speech-conditioned robot manipulation) or text outputs (context-grounded VQA). Moreover, Unified-IO 2 is trained mainly on simple sound-event tasks and lacks speech understanding, treating speech only as a simple audio event "there's someone speaking", making dialogue and instruction following impossible.
>
> **Gato** and **PaLM-E** both lack end-to-end speech understanding and generation. In addition, when dealing with action generation, PaLM-E outputs textual descriptions which need further explanation rather than precise actions, whereas ELLSA directly produces actions, speech, or text in a unified framework. These distinctions highlight ELLSA’s unique focus on integrated, real-time multimodal understanding and generation.
>
> ---
>
> **Weakness 2: Limited Technical Novelty of SA-MoE: While effective, SA-MoE is essentially routing + shared attention between existing pretrained models.**
>
> We thank the reviewer for acknowledging that “SA-MoE is an elegant solution that leverages pretrained experts while mitigating modality interference”. We also understand the concern that SA-MoE may resemble prior work, and here we clarify its conceptual and technical contributions.
>
> Although SA-MoE is intentionally simple, it represents one of the **earliest attempts to fuse concurrent multimodal input and output streams** involving speech, vision, text, and action. Previous multimodal MoE studies have largely focused on single-modality understanding (e.g., vision or audio), where MoE modules are primarily used to mitigate domain conflicts. However, when extending MoE to complex multimodal integration, as in our setting involving speech, vision, text, and action, new challenges arise. On one hand, training such a system from scratch requires substantial amounts of multimodal data. The distribution of these data is often highly imbalanced across modalities, necessitating carefully designed training strategies to ensure fair and effective cross-modal learning. On the other hand, many powerful pretrained models already exist for subsets of modalities, such as speech–text models (our speech expert) or vision–text–action models (our action expert). These models are trained on large, modality-specific datasets and exhibit strong capabilities. Efficiently leveraging these pretrained experts would significantly reduce training costs and improve performance for a unified multimodal model.
>
> However, existing MoE architectures are designed primarily as pretraining frameworks, and thus cannot be directly used to connect multiple pretrained experts. Because their MoE modules operate only within the FFN layers, they are unable to accommodate the distinct sets of attention parameters from different experts. To address this gap, we introduce SA-MoE, a novel and data-efficient approach for fusing pretrained experts while effectively mitigating modality interference through attention interaction. In summary, while prior MoE approaches mainly provide strong **pretraining** architectures, SA-MoE offers a high-performance, data-efficient architecture for **post-training** multimodal expert fusion. These discussions are also provided in Appendix F.

---

> ### Author Response · Authors · 2025-11-21
> **Response to Reviewer 4T5n (part2)**
>
> **Weakness 3: Speaking-while-acting shows notable performance drops (Table 4), especially on harder tasks (LIBERO LONG: 73.2% vs 84.4%)**
>
> Thanks for pointing out this weakness of our model. We first emphasise that **speaking-while-acting is inherently more challenging than speaking or acting alone.** In this setting, the model must perform two tasks simultaneously, which naturally introduces competition for attention and potential distraction, as discussed in Section 5.2.2. Therefore, a performance drop relative to single-modality tasks is expected.
>
> Second, our analysis indicates that the drop is largely attributable to **the capacity of the underlying components**, rather than a limitation of the overall architecture. To verify this, we constructed an enhanced version of ELLSA by replacing its speech encoder with SPEAR [1], a substantially stronger zipformer encoder trained on larger and more diverse datasets. With this upgrade, ELLSA not only improves in its base capabilities but also **substantially reduces the speaking-while-acting gap**: from 13.3% to 2.9% on LIBERO LONG, and from 17.0% to 7.0% on web_questions. This supports our claim that higher-capacity backbone components can effectively mitigate the degradation in the multi-task, concurrent-stream setting.
>
> [1] Yang, Xiaoyu, et al. "SPEAR: A Unified SSL Framework for Learning Speech and Audio Representations." arXiv preprint arXiv:2510.25955 (2025). https://huggingface.co/collections/marcoyang/spear-encoders
>
> ### Ablation study on speech encoder.
>
> **(a) Speech interaction S2T performance.**
>
> | Speech Encoder |  Speaking alone | Llama Q. | Web Q. | TriviaQA  | AlpacaE. | Speaking-while-acting | Llama Q. | Web Q. | TriviaQA  | AlpacaE. |
> | :--- | :--- | :--- | :--- | :--- | :--- | :--- | :--- | :--- | :--- | :--- |
> | mamba | | 74.7 | 39.5 | 45.2 | 3.09 | | 68.9 (-7.8%) | 32.8 (-17.0%) | 35.1 (-22.3%) | 2.66 (-13.9%) |
> | SPEAR | | 76.7 | 48.7 | 61.5 | 3.81 | | 74.8 (-2.5%) | 45.3 (-7.0%) | 53.2 (-13.5%) | 3.67 (-3.7%) |
>
> **(b) Robot manipulation performance.**
>
> | Speech Encoder |  Acting alone | SPATIAL | OBJECT | GOAL | LONG | Speaking-while-acting | SPATIAL | OBJECT  | GOAL  | LONG |
> | :--- | :--- | :--- | :--- | :--- | :--- | :--- | :--- | :--- | :--- | :--- |
> | mamba | | 90.8% | 95.8% | 86.4% | 84.4% | | 93.3% (+2.8%) | 96.6% (+0.8%) | 86.1% (-0.3%) | 73.2% (-13.3%) |
> | SPEAR | | 91.2% | 96.6% | 90.4% | 88.8% | | 90.9% (-0.3%) | 96.6% (-0.0%) | 89.7% (-0.8%) | 86.2% (-2.9%) |
>
> ---
>
> **Question 1: Why specifically 2 experts rather than 4 (one per modality)? Have you experimented with different expert configurations? What is the computational overhead of SA-MoE vs a single model**
>
> Thank you for the insightful question. We choose to include two experts in ELLSA because the speech expert is trained on both **speech+text**, whereas the action expert is trained on both **vision+action**. Grouping speech with text and vision with action allows us to most effectively leverage the pretrained capabilities of each model, as described in Section 3.2.
>
> We also conducted ablations on the number of experts. Due to memory constraints, a full 4-expert MoE is not feasible, so we evaluated two 3-expert variants by splitting either (1) vision/action or (2) speech/text into separate experts (with split experts initialized from the same pretrained model). Both 3-expert configurations perform comparably to the 2-expert SA-MoE, and given the additional efficiency of fewer experts, we adopt the 2-expert design. These discussions are now presented in Appendix G (Number of experts).
>
> Regarding computational overhead, each training step of SA-MoE is indeed more expensive than that of a single model. However, **the overall training cost is significantly lower**, because SA-MoE requires far fewer training steps to reach strong performance, as demonstrated in Appendix F. For example, SA-MoE trained for only 500 steps can surpass a single model trained for 3000 steps, thanks to its effective reuse of pretrained experts. At inference time, SA-MoE introduces **no additional cost**, since only the selected expert’s parameters are active at any given moment, as explained in Section 3.2.
>
> ### Ablation study on the number of experts.
>
> **(a) Speech interaction S2T performance.**
>
> | Number of experts | Llama Q. | Web Q. | TriviaQA | AlpacaEval |
> | :--- | :--- | :--- | :--- | :--- |
> | 2 (Speech, Action) | 74.7 | 39.5 | 45.2 | 3.09 |
> | 3 (Speech, Vision, Action) | 70.3 | 35.9 | 45.9 | 3.08 |
> | 3 (Speech, Text, Action) | 73.0 | 34.7 | 45.0 | 2.93 |
>
> **(b) Robot manipulation performance.**
>
> | Number of experts | SPATIAL | OBJECT | GOAL | LONG |
> | :--- | :--- | :--- | :--- | :--- |
> | 2 (Speech, Action) | 90.8% | 95.8% | 86.4% | 84.4% |
> | 3 (Speech, Vision, Action) | 89.8% | 96.2% | 85.8% | 84.4% |
> | 3 (Speech, Text, Action) | 89.0% | 97.4% | 87.2% | 87.8% |

---

> ### Author Response · Authors · 2025-11-21
> **Response to Reviewer 4T5n (part3)**
>
> **Question 2: Can the model handle truly simultaneous inputs (e.g., receiving new speech while still processing previous speech and executing action)? The 1-second time block seems to discretize the interaction.**
>
> Thank you for the thoughtful question. This indeed touches on the essence of full-duplex multimodal interaction. We use a 1-second time block because the action expert, UniVLA, produces 10 action frames per second; aligning with this rate allows us to fully leverage its pretrained capabilities.
>
> It is important to clarify that **no** full-duplex model can accept input at arbitrarily fine granularity during ongoing generation. Full-duplex processing necessarily operates on **discrete time intervals**, since LLM inference itself is discrete. Thus, "full-duplex" always refers to simultaneous input–output behavior at a particular time scale. The key distinction relative to turn-based models is that **full-duplex models explicitly model *time***: during each block, the model both receives new multimodal inputs and produces outputs. In contrast, turn-based models, without *time* modeling, must finish receiving all inputs before generation begins and cannot process additional input until generation completes. By modeling time explicitly, full-duplex models can internally manage natural interactive dynamics, such as turn-taking (e.g. When to start speaking) and interruptions, capabilities that turn-based models fundamentally lack.
>
> Although our 1-second block reflects a practical compromise to integrate the action expert, it already enables ELLSA to process **truly concurrent** inputs and outputs: the model receives new speech and visual signals while simultaneously generating speech and action predictions at every time step. To further demonstrate flexibility of our framework, we implemented a variant with a finer 0.48-second block, generating 4 text tokens and 5 action frames per cycle. The performance drop of action expert is likely due to the shorter action sequences, which can reduce the temporal coherence of the generated actions, maybe a constraint of original action expert. As shown below, the model continues to operate effectively at this finer temporal resolution.
>
> ### Ablation study on time block duration.
>
> **(a) Speech expert performance.**
>
> | Time block | Llama Q. | Web Q. | TriviaQA | AlpacaEval |
> | :--- | :--- | :--- | :--- | :--- |
> | 1s | 77.7 | 44.1 | 55.9 | 3.35 |
> | 0.48s | 78.5 | 44.8 | 55.9 | 3.68 |
>
> **(b) Action expert performance.**
>
> | Time block | SPATIAL | OBJECT | GOAL | LONG |
> | :--- | :--- | :--- | :--- | :--- |
> | 1s | 95.4% | 98.8% | 93.6% | 94.0% |
> | 0.48s | 91.0% | 92.4% | 84.2% | 81.0% |
>
> **(c) SA-MoE performance.**
>
> | Time block | Llama Q. | Web Q. | TriviaQA | AlpacaE. | SPATIAL | OBJECT | GOAL | LONG |
> | :--- | :--- | :--- | :--- | :--- | :--- | :--- | :--- | :--- |
> | 1s | 74.7 | 39.5 | 45.2 | 3.09 | 90.8% | 95.8% | 86.4% | 84.4% |
> | 0.48s | 71.7 | 38.4 | 44.7 | 2.95 | 81.0% | 85.0% | 73.4% | 71.6% |
>
> **(d) Per-time-block latency.**
>
> | Time block | Speech-to-speech | Speech-to-action |
> | :--- | :--- | :--- |
> | 1s | 854ms | 786ms |
> | 0.48s | 455ms | 428ms |
>
> ---
>
> **Question 3: What are the main challenges anticipated for real-world deployment? Have you done any preliminary real robot experiments?**
>
> We sincerely appreciate the reviewer’s suggestion to evaluate the system in real-world robotic settings. While our primary focus in this paper is on human-like, real-time multimodal interactive agent models, deploying the system on physical robotic hands is beyond the current scope of our work. That said, we would like to emphasise that this limitation is practical rather than algorithmic. There are no methodological barriers that would prevent our approach from being applied to real robots. As an academic lab without access to a full robotic manipulation platform, conducting such experiments at this stage is challenging for us. Nevertheless, we believe our method is well-positioned for real-world deployment for several reasons:
> - **Sim-to-real evidence**: Prior work on Vision-Language-Action (VLA) models (e.g., π-0, OpenVLA) demonstrates strong sim-to-real transfer after fine-tuning with even modest amounts of real robot data.
> - **SA-MoE architectural design**: The SA-MoE framework integrates pretrained speech and action experts while preserving their specialised strengths. Substituting our current action expert with one finetuned on physical robot data should enable effective operation on real hardware.
> - **Real-time performance**: Our latency analysis shows that the system meets real-time inference requirements, which is critical for physical control.
>
> In summary, although current resource constraints prevent us from performing real-robot evaluations, we do not anticipate any fundamental obstacles to deploying our approach on real robotic systems. We appreciate the reviewer’s understanding of these practical limitations.

---

> > ### Author Response · Authors · 2025-11-27
> > **Request of Reviewer's attention and feedback**
> >
> > Dear Reviewer 4T5n,
> >
> > As a gentle reminder, it has been more than six days since we submitted our rebuttal, and we wanted to kindly check whether our responses have addressed your main concerns. If anything remains unclear or you would like further clarification or experiments, we would be very happy to provide them within the remaining discussion time.
> >
> > Thank you again for your valuable and constructive feedback, which has inspired further improvements to our paper.
> >
> > Best regards,
> >
> > The Authors

---

### Author Response · Authors · 2025-12-03
**Summary of Rebuttal and Contributions**

We thank all reviewers, the AC, and the SAC for their thoughtful feedback. Below, we provide a concise overview of our contributions and how we addressed the major concerns raised during the rebuttal period.

---

### Key Contributions

- **ELLSA is the first end-to-end full-duplex multimodal model** that performs simultaneous perception and generation across vision, text, speech, and action, going beyond all prior turn-based or modality-isolated systems. All reviewers acknowledge the motivation and novelty of our paper.
- **SA-MoE introduces a clean, scalable, and data-efficient fusion architecture** that integrates pretrained modality experts while explicitly resolving modality interference; three reviewers (4T5n,9jnm and yfkh) recognised it as an elegant and technically well-motivated design.
- **ELLSA demonstrates capabilities not seen in prior work**, including speaking-while-acting, action barge-ins, turn-taking, and context-grounded VQA during manipulation, while maintaining similar performance with modality-specific baselines, offering strong evidence of more natural and human-like multimodal interaction. All reviewers acknowledge the experimental results.

---

### Summary of major reviewer concerns and our responses

During the rebuttal period, we tried our best to address all reviewers' concerns. We also revise the PDF according to the suggestions from all reviewers. All revisions in the updated PDF are marked in blue. Unfortunately, none of the reviewers engaged in the discussion.

Across all reviewers, we clarified ELLSA’s distinctions from prior works, explained SA-MoE’s design insights, clarified the full-duplex concept, refined claims (including removing overstated AGI language), and added experiments: enhanced speech encoder, expert-number ablations, finer time-block variants, latency measurements, and CALVIN generalization. Here we would like to summarize the reviewers' common concerns.

- **Technical novelty of SA-MoE (4T5n, smUE).** We clarify that unlike prior MoE work focusing on pretraining, SA-MoE provides a post-training high-performance fusion architecture, introducing bidirectional cross-expert attention over interleaved multimodal sequences.
- **Speaking-while-acting performance drop (4T5n, yfkh).** We show the drop stems from component capacity, not architectural limits. Using a stronger speech encoder (SPEAR) greatly reduces the gap (details in Appendix G).
- **Number of experts (4T5n, smUE).** Two 3-expert variants perform similarly to the 2-expert version, while the latter is more efficient. We therefore retain the 2-expert design (Appendix G).
- **Effect of time block size (4T5n, 9jnm, smUE).** We added new results using a finer 0.48-second block, which preserves speech quality but degrades action performance due to reduced temporal coherence. A 1-second block offers the best overall trade-off (Appendix G).
- **Generalization and real-world deployment (4T5n, 9jnm, smUE).** New experiments on the CALVIN environment show that SA-MoE naturally generalizes by substituting task-specific action experts, indicating a straightforward path toward real-robot deployment (Appendix H).
- **Latency (yfkh, smUE).** Speech→speech and speech→action latencies (854 ms / 786 ms on an A100) validate the feasibility of real-time full-duplex streaming.

Finally, we sincerely thank all reviewers and the AC for their time and feedback. We humbly believe that our rebuttal and new results can adequately address all concerns and substantially strengthen the paper.

---

### Meta-Review · Area_Chair_y2LK · 2026-01-08

**Summary:**

The reviewers recognized the significant ambition and timely nature of this work, which introduces ELLSA, the first end-to-end, full-duplex model capable of simultaneous perception and generation across vision, speech, text, and action modalities. While initial evaluations were polarized (scores of 8, 6, 4, 4), the collective concerns centered on three primary areas:

- Technical Novelty: The distinction between the proposed SA-MoE (Self-Attention Mixture-of-Experts) and existing MoE architectures used for multmodal learning.
- Concurrent Performance: The observed performance degradation in "speaking-while-acting" tasks compared to single-modality baselines.
- Generalization: The model's ability to generalize beyond the LIBERO benchmark to other environments or real-world settings, especially non-simulated robot environments.

Authors rebuttal successfully addresed most of concerns despite one outstanding concern about sim-to-real gap. Given the timely nature and promising potential of this work, the meta-review feels the outstanding concern will not be a significant gap. I expect this work to spur more works along this direction to enable truly interactive physical AIs.

**Reviewer Concerns:**

Reviewers had the following concerns that are addresed in rebuttal:

- architecture novelty: The authors clarified that unlike prior MoE research focused on pretraining (e.g., EVE-v2, Mono-InternVL), SA-MoE is a post-training fusion architecture. It provides a light-weighted way to unite pretrained models from different modalities.


- performance drops in concurrent tasks: To address the "distraction" or performance drop during simultaneous tasks, the authors provided new experiments using a stronger speech encoder (SPEAR). These results demonstrated that the performance gap (e.g., on LIBERO LONG) was reduced from 13.3% to 2.9%, suggesting the limitation is component capacity rather than the SA-MoE architecture itself.

- generalization to new tasks: The authors added results from the CALVIN benchmark, showing that the SA-MoE framework generalizes naturally by substituting task-specific action experts.

- runtime complexity and latency: Empirical measurements were provided showing speech-to-speech and speech-to-action latencies of 854 ms and 786 ms, respectively, confirming the model functions within its 1-second time block.

- validity of the "full-duplex" definition: the author explained the necessity of discretization in full duplex systems with LLM and provided experiments with different time block sizes.

The following concern is still outstanding afer rebuttal:

- Real-World Deployment: While authors argued for the feasibility of sim-to-real transfer based on prior VLA literature, the model has yet to be validated on physical hardware. However, the authors noted this as a practical resource limitation rather than a methodological one. The meta-reviewer agree with this clarification but still encourage the authors to seek oppurtunity for real-world experiments to amplify the impact of this work.

**Reviewer Scores:**

Reviewer 4T5n (Initial Score: 4): This reviewer is likely to raise their score to a 6. The authors addressed the primary technical concerns by implementing new ablations on the number of experts and time block durations (1s vs 0.48s). Additionally, the authors clarified the novelty of SA-MoE as a post-training fusion framework and cited the previously omitted prior works.

Reviewer 9jnm (Initial Score: 8): This reviewer is expected to maintain their score of 8.

Reviewer yfkh (Initial Score: 6): This reviewer is likely to keep their initial rating.

Reviewer smUE (Initial Score: 4): This reviewer is expected to raise their score to a 5 or 6. The authors addressed the concern regarding limited generalizability by providing new results from the CALVIN benchmark, which showed that the SA-MoE framework extends to different VLA environments. They also provided empirical latency measurements to address concerns regarding real-time constraints.

---

### Decision · Program_Chairs · 2026-01-26

Accept (Poster)